# Privacy-Preserving Reinforcement Learning from Human Feedback via Decoupled Reward Modeling

### Abstract

Preference-based fine-tuning has become an important component in training large language models, and the data used at this stage may contain sensitive user information. A central question is how to design a differentially private pipeline that is well suited to the distinct structure of reinforcement learning from human feedback. We propose a privacy-preserving framework that imposes differential privacy only on reward learning and derives the final policy from the resulting private reward model. Theoretically, we study the suboptimality gap and show that privacy contributes an additional additive term beyond the usual non-private statistical error. We also establish a minimax lower bound and show that the dominant term changes with sample size and privacy level, which in turn characterizes regimes in which the upper bound is rate-optimal up to logarithmic factors. Empirically, synthetic experiments confirm the scaling predicted by the theory, and experiments on the Anthropic HH-RLHF dataset using the Gemma-2B-IT model show that the proposed private reward-learning method outperforms the private baselines considered and approaches the non-private reward model as the privacy budget is relaxed.

**Keywords:** Large Language Models, Preference Fine-tuning, Differential Privacy, Reward Modeling, Sample Complexity.

**Mathematics Subject Classification (2020):** 62XXX

## 1 Introduction

Large language models (LLMs) are increasingly used as general-purpose tools across a growing range of downstream and domain-specific applications, including medicine, finance, law, and science (Nie et al., 2024; Qin and Sun, 2024; Zheng et al., 2025; Zhou et al., 2026). In practice, adapting a pretrained model to such settings typically relies on post-training or fine-tuning, which has become a standard part of modern LLM deployment (Ouyang et al., 2022; Bai et al., 2022; Chia et al., 2025). Preference-based LLM fine-tuning is one widely used form of this adaptation. Such fine-tuning is commonly carried out through reinforcement learning from human feedback (RLHF) (Christiano et al., 2017; Stiennon et al., 2020; Ouyang et al., 2022), where comparative feedback is used to improve a policy when reward feedback is not directly observed.

In particular, a typical LLM alignment pipeline begins with a pretrained model, obtains a reference policy through supervised fine-tuning (SFT) on expert-written or carefully curated instruction-response pairs, and then refines that policy using pairwise preferences over candidate responses, typically through RLHF. Because SFT data are costly to scale and may not

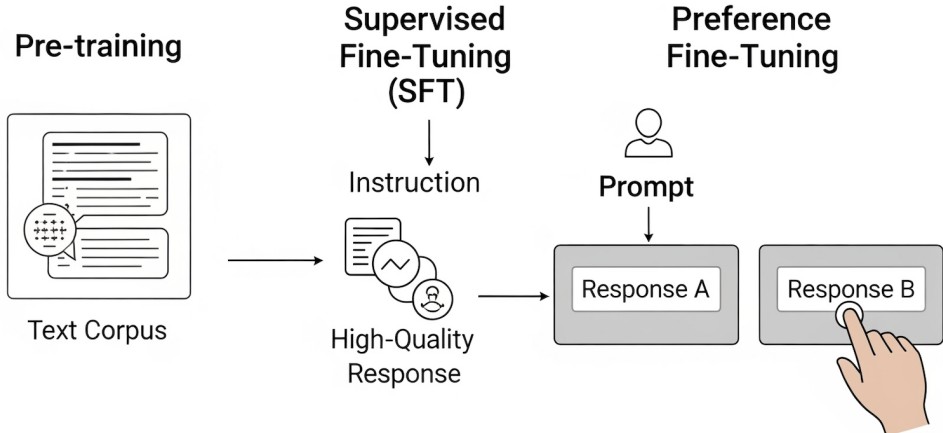

Figure 1: A typical large language model adaptation pipeline. We focus on privacy during the preference fine-tuning stage, where sensitive user interactions can be directly reflected in training records.

fully capture the nuanced judgments needed for alignment, preference-based fine-tuning has become a common extension beyond SFT (Chia et al., 2025). Figure 1 illustrates this canonical RLHF pipeline. Modern preference-based alignment also includes direct formulations such as direct preference optimization (DPO), which bypass explicit reward-model training and instead optimize the policy directly from preference data (Rafailov et al., 2024).

On the other hand, this preference-based fine-tuning pipeline can raise privacy concerns. Modern models are susceptible to training-data extraction attacks (Carlini et al., 2021; Nasr et al., 2023) and membership inference attacks (Shokri et al., 2017; Wu and Cao, 2025), which can reveal whether a user's data was used for training and, in some cases, expose training examples. In preference-based LLM fine-tuning beyond SFT, the training signal is not merely a scalar label, but a full interaction tuple $(x_i, a_i^1, a_i^2, y_i)$, where $x_i$ is the user prompt, $(a_i^1, a_i^2)$ are candidate responses, and $y_i$ is the associated preference label. Even when explicit identifiers are absent, the prompt $x_i$ may contain sensitive or potentially identifying context, and the derived responses and label may also reveal private user information. Figure 2 illustrates such an example. These concerns call for principled protection against leakage at the tuple level.

Differential privacy (DP) (Dwork et al., 2006) provides a widely used formal framework for privacy-preserving data analysis and machine learning. A central question is what object should be protected by the privacy definition. Much of the existing literature adopts label-DP (Chowdhury et al., 2024; Zhang et al., 2025; Teku et al., 2025; Wu et al., 2025), which protects only the preference label and is therefore most naturally aligned with protecting the privacy of the preference-label annotator. Our focus is different. In the settings we study, the relevant target is the end-user, so privacy should apply to the full interaction tuple rather than to $y_i$ alone. This motivates tuple-level privacy for $(x_i, a_i^1, a_i^2, y_i)$.

A key distinction from standard RL is that RLHF does not receive rewards directly from the environment. Instead, it must infer a latent reward signal from preference data, which introduces an additional reward-learning layer into the pipeline. Under DP, this distinction is consequential. Privacy can now enter at reward estimation, at downstream policy optimization, or at both stages. If privacy is imposed naively on such a multi-stage pipeline, noise can accumulate across

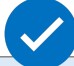

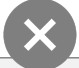

I'm a 35-year-old software engineer living in the Bay Area. I was diagnosed with early-onset Parkinson's two years ago. My main symptoms are hand tremors and sleep disturbances. My neurologist recently suggested switching from Azilect to a dopamine agonist, but I'm concerned about the side effects like impulse control problems I've read about online. What is the typical success rate for this switch in patients with my profile?

For patients in their 30s with early-onset Parkinson's, switching from an MAO-B inhibitor like Azilect to a dopamine agonist is a common strategy to manage motor symptoms. While success rates vary, many find significant relief. The concerns about impulse control disorders are valid, affecting a minority of patients, but this risk can be monitored closely with your neurologist. Regular follow-ups to screen for these behavioral changes are a standard part of this treatment path.

Decisions about changing medication for Parkinson's disease should be made in careful consultation with your neurologist. They can provide information on the risks and benefits of different treatments based on your specific health profile and help you make an informed choice.

Figure 2: A sensitive interaction record. Even without direct identifiers, prompts can contain quasi-identifiers whose combination may re-identify an individual, motivating tuple-level protection of $(x_i, a_i^1, a_i^2, y_i)$.

stages, or the data or privacy budget must be divided across them. In this sense, the reward-learning layer in RLHF is not only a modeling feature but also a central channel through which privacy constraints affect utility.

This raises a second central question beyond what should be protected. One must also decide where privacy should enter the alignment pipeline so that downstream utility is preserved as much as possible. Existing directions already illustrate several possibilities, including introducing DP to DPO (Chen et al., 2025) and introducing DP to each stage of RLHF (Wu et al., 2024). In this sense, existing methods largely arise by applying DP to existing non-private alignment pipelines. The remaining question is whether RLHF admits a privacy-aware design that uses its own structure more directly.

In this work, we take a different route. Rather than adding DP to each step of an existing alignment pipeline, we use the two-stage structure of RLHF itself to guide where privacy should enter. We develop a tuple-level private framework that places privacy on reward learning and treats downstream policy improvement as post-processing. We then study the resulting privacy–utility tradeoff through theory and numerical experiments.

## 1.1 Our Contributions

Our contributions are summarized as follows.

- **Methodology.** We propose a private RLHF framework that exploits a structural feature specific to RLHF, rather than simply adding DP to an existing alignment pipeline. Unlike standard RL, where the environment directly provides reward, RLHF first uses preference data to train a reward model and then updates a reference policy using that learned reward. Our framework places DP only on the reward-learning stage and treats downstream policy as post-processing of the resulting private reward model. Placing privacy on reward learning creates a buffer between DP noise and the downstream policy. In LLM fine-tuning, this avoids privatizing the policy itself, which can yield responses worse than the reference

policy, and instead uses the private reward model only to re-rank reference-generated responses, without additional privacy cost.

- **Theory.** We study the suboptimality gap of the policy induced by our framework and show that privacy enters as an additional additive term on top of the usual non-private statistical error. We also establish, to the best of our knowledge, the first minimax lower bound for this private RLHF problem. A key difficulty is that the dominant lower-bound term changes with the balance between sample size and privacy budget. We therefore characterize the regimes in which our method is rate-optimal up to logarithmic factors, including the case where the sample size is sufficiently large relative to the privacy scale for the optimal privacy-dominated rate to emerge.

- **Empirical validation.** We validate the framework on synthetic preference-learning experiments and on an LLM alignment task based on the Anthropic HH-RLHF dataset (Bai et al., 2022) using the Gemma-2B-IT model (Team et al., 2024). The synthetic results confirm the theoretically predicted scaling with sample size, privacy budget, and dimension, and show that our method substantially reduces the incidence of policies that underperform the reference policy relative to private policy-optimization baselines. In the LLM experiment, our method attains the highest accuracy in ranking the preferred response above the rejected response among the private methods considered, and its accuracy approaches that of the non-private reward model as the privacy budget is relaxed.

## 1.2 Related Work

We briefly review prior work on preference-based policy learning and its theoretical foundations, as well as DP for alignment.

**Preference-based policy learning and KL-regularized preference optimization.** One of the most widely used applications of RLHF is preference fine-tuning for LLMs (Christiano et al., 2017; Stiennon et al., 2020; Ouyang et al., 2022; Bai et al., 2022; Ye et al., 2025). More recently, direct alignment methods have gained prominence by optimizing a closed-form objective that avoids explicit reward-model training and RL in the loop (Rafailov et al., 2024; Garg et al., 2025). From a theoretical viewpoint, a growing line of work develops finite-sample guarantees of KL-regularized preference optimization and iterative procedures, clarifying when logged preference data suffices for reliable policy improvement (Xiong et al., 2024; Ye et al., 2024; Xiong et al., 2024; Song et al., 2024; Zhao et al., 2024). Several recent works also revisit the role of the KL regularizer itself—including its interpretation, limitations, and variants (Huang et al., 2024; Aminian et al., 2025; Liu et al., 2025a; Li et al., 2026). Our work complements this theory by quantifying how DP affects the KL-regularized preference optimization.

**DP and privacy-preserving alignment.** DP is a *de facto* standard in privacy-preserving framework in machine learning (Dwork et al., 2006). A large literature studies optimal accuracy guarantees under DP for empirical risk minimization and stochastic convex optimization (Dwork et al., 2014; Chaudhuri et al., 2011; Kifer et al., 2012; Bassily et al., 2014; Wang et al., 2017; Bassily et al., 2019). Among many, the most widely used optimization method is noisy gradient-based training, popularized by DP-SGD and closely related mechanisms (Dwork et al.,

2006; Abadi et al., 2016; Bu et al., 2020). This line of work has also motivated refined privacy accounting frameworks and composition results that tighten end-to-end privacy loss in iterative training (Kairouz et al., 2015; Mironov, 2017; Bun and Steinke, 2016; Dong et al., 2022).

Building on these foundations, recent work has begun to study DP for preference-based alignment and RLHF. A first key distinction concerns the unit of protection. Much of the recent literature studies label-DP, which protects only the preference label while treating the prompt and candidate responses as public or non-sensitive. This formulation is naturally aligned with protecting annotator feedback, but it does not address leakage from the text content itself. This perspective underlies several recent works on private DPO and related offline alignment objectives (Zhang et al., 2025; Zhou et al., 2025a,b; Teku et al., 2025). In contrast, our focus is tuple-level privacy, where the prompt, candidate responses, and preference outcome may all carry sensitive information.

A second distinction concerns how privacy is incorporated into the alignment pipeline. Most existing approaches begin with a non-private alignment framework and then introduce a DP mechanism on top of it. When the target is label-DP, a standard way to privatize the labels is randomized response (Warner, 1965), and several recent works follow this route (Zhang et al., 2025; Zhou et al., 2025a,b; Teku et al., 2025). When the target is tuple-level DP, privacy is typically enforced through noisy gradient-based training such as DP-SGD, including private adaptations of DPO and related RLHF procedures (Chen et al., 2025; Wu et al., 2024, 2025; Chowdhury et al., 2024; Korkmaz and Brown-Cohen, 2024). Our contribution differs from these directions both methodologically and theoretically. We study a framework tailored to the distinct structure of RLHF, with privacy imposed only on reward learning, and we analyze the resulting suboptimality gap together with minimax lower bounds that characterize when the corresponding rates are optimal up to logarithmic factors.

## 1.3 Paper organization and notation

The remainder of the paper is organized as follows. Section 2 introduces the preliminaries for our work, including DP and RLHF. Section 3 presents the proposed decoupled framework. In section 4, we develop the theoretical analysis, deriving both upper and lower bounds on the suboptimality gap of the induced policy and identifying regimes in which these bounds match up to logarithmic factors. Section 5 reports numerical studies on both synthetic examples and LLM fine-tuning experiments. Section 6 concludes with a discussion of the main implications, limitations, and directions for future work. Most proofs are deferred to the Supplementary Material Section D, while straightforward arguments are given in the main text.

For notation, we use standard asymptotic order notation. For two positive sequences $a_n$ and $b_n$, $a_n = O(b_n)$ means that $a_n/b_n$ is uniformly bounded, and $a_n = \Omega(b_n)$ means that $a_n/b_n$ is bounded away from zero. We write $a_n = \widetilde{O}(b_n)$ and $a_n = \widetilde{\Omega}(b_n)$ when the corresponding relations hold up to polylogarithmic factors.

# 2 Preliminaries

In this section, we establish the background for our work. We begin by formally outlining the standard pipelines for preference fine-tuning, then introduce the background on DP.

## 2.1 Reinforcement Learning from Human Feedback

A widely used template for learning from preferences is KL-regularized reinforcement learning from human feedback (RLHF). The key object is a reference policy $\pi_0$, and the goal is to improve decision making while preventing the updated policy from drifting too far from $\pi_0$ or overfitting to a limited preference dataset. This viewpoint is especially prominent in large language model alignment, where preference fine-tuning is a standard stage after a strong base policy is obtained.

We consider an offline preference dataset originally recorded as $\mathcal{D} = \{(x_i, a_i^1, a_i^2, y_i)\}_{i=1}^n$, where $x_i \in \mathcal{X}$ denotes a context, such as a prompt in LLM applications, $a_i^1, a_i^2 \in \mathcal{A}$ are two candidate actions, and $y_i$ indicates which candidate is preferred. For the theoretical development, it is more convenient to rewrite each record in ordered form as $(x_i, a_i^w, a_i^l)$, where $a_i^w \in \mathcal{A}$ and $a_i^l \in \mathcal{A}$ denote the preferred and non-preferred actions, respectively. A conventional modeling assumption is then the Bradley–Terry preference model (Bradley and Terry, 1952).

**Definition 1** (Bradley–Terry Model)**.** *For a context $x$ and a preferred/non-preferred pair $(a^w, a^l)$, the probability that $a^w$ is preferred to $a^l$ is modeled as*

$$\mathbb{P}(a^w \succ a^l \mid x) = \frac{\exp(r^*(x, a^w))}{\exp(r^*(x, a^w)) + \exp(r^*(x, a^l))} = \sigma(r^*(x, a^w) - r^*(x, a^l)),$$

*where $\sigma(t) = (1 + e^{-t})^{-1}$ is the sigmoid function.*

In KL-regularized RLHF, the reference policy $\pi_0$ is treated as a strong baseline that encodes prior knowledge and safe behavior, and KL regularization controls the magnitude of the preference-driven update. In the LLM preference fine-tuning pipeline, a common choice of $\pi_0$ is a model that has already been trained on a large supervised instruction-following dataset, often referred to as supervised fine-tuning, or SFT (Ouyang et al., 2022). This choice reflects the practical role of $\pi_0$ as an anchor to a broadly competent response distribution, while preference data provide an additional signal that refines behavior without requiring the policy to relearn basic capabilities from scratch.

Under Definition 1, the standard reward-modeling phase estimates a parametric reward function $r_\theta \in \mathcal{R} = \{r_\theta : \theta \in \Theta\}$ by maximizing the log-likelihood

$$\hat{\theta} = \operatorname*{argmax}_{\theta \in \Theta} \sum_{i=1}^n \log \sigma \left( r_\theta(x_i, a_i^w) - r_\theta(x_i, a_i^l) \right). \tag{1}$$

Once a reward estimator $\hat{r}$ is obtained, the goal is to derive a policy that achieves high reward while remaining close to a reference policy $\pi_0$, which is often the SFT model in LLM pipelines. For any reward function $r$ on $\mathcal{X} \times \mathcal{A}$, we evaluate a policy $\pi$ by the KL-regularized value

$$V_\eta(\pi; r) = \mathbb{E}_{x \sim d_0} \left[ \mathbb{E}_{a \sim \pi(\cdot | x)} \big[ r(x, a) \big] - \frac{1}{\eta} \mathrm{KL} \big( \pi(\cdot \mid x) \,\|\, \pi_0(\cdot \mid x) \big) \right], \tag{2}$$

where $d_0$ denotes the context distribution, and $\eta > 0$ controls the strength of regularization. Let $\pi_r^\eta \in \arg\max_\pi V_\eta(\pi; r)$ denote the KL-regularized optimizer under $r$. The closed form solution is as follows:

**Lemma 2** (Policy Improvement Oracle). *For a fixed context $x$, reward $r$, and reference policy $\pi_0$, any maximizer $\pi_r^\eta \in \arg\max_\pi V_\eta(\pi; r)$ satisfies*

$$\pi_r^\eta(a \mid x) = \frac{1}{Z_r(x)} \pi_0(a \mid x) \exp(\eta\, r(x,a)), \tag{3}$$

*where $Z_r(x) = \mathbb{E}_{a\sim\pi_0(\cdot\mid x)}\big[\exp\big(\eta r(x,a)\big)\big]$.*

Although $\pi_r^\eta$ admits a closed-form expression, exact sampling from this policy is typically infeasible when the action space is large. The expression involves a normalization term that aggregates $\exp(\eta r(x,a))$ over the action space, which is computationally prohibitive in LLM settings where actions correspond to long token sequences. This motivates practical methods that approximate the KL-regularized optimizer through parameterized learning.

A common approach is to optimize a parameterized policy with reinforcement learning algorithms such as proximal policy optimization (PPO) (Schulman et al., 2017), which targets the KL-regularized objective while avoiding explicit normalization over $\mathcal{A}$. Since PPO does not appear explicitly in the main development, we defer a brief background discussion to Appendix A.

An alternative is Direct Preference Optimization (DPO) (Rafailov et al., 2024), which leverages the identity implied by (3). In particular, for $\pi_r^\eta$ one can write

$$r(x,a) = \frac{1}{\eta} \log \frac{\pi_r^\eta(a \mid x)}{\pi_0(a \mid x)} + \frac{1}{\eta} \log Z_r(x).$$

Substituting this relation into the pairwise preference likelihood (1) eliminates the unknown $Z_r(x)$ and yields a supervised objective over policy parameters

$$\mathcal{L}_{\text{DPO}}(\theta; \mathcal{D}, \pi_0) = - \sum_{(x,a^w,a^l)\in\mathcal{D}} \log \sigma \left( \frac{1}{\eta} \log \frac{\pi_\theta(a^w \mid x)}{\pi_0(a^w \mid x)} - \frac{1}{\eta} \log \frac{\pi_\theta(a^l \mid x)}{\pi_0(a^l \mid x)} \right). \tag{4}$$

This derivation connects DPO with the Bradley–Terry model. In particular, Lemma 2 implies that the normalizing term $Z_r(x)$ cancels in pairwise comparisons, so under the reward-induced ratio class, the pairwise DPO logit in (4) coincides with the Bradley–Terry reward difference $r_\theta(x, a^w) - r_\theta(x, a^l)$. In this sense, DPO represents the reward model implicitly through the policy (Rafailov et al., 2024).

The practical appeal of DPO and its variants (Azar et al., 2024; Ethayarajh et al., 2024; Xu et al., 2024; Meng et al., 2024) is that preference optimization becomes a single-stage supervised learning problem, without a separate reward learning, iterative RL sampling, or the sensitive tuning choices in RL-based policy optimization. Under DP, however, this change of variables also changes the object to which the privacy mechanism is applied. Directly privatizing (4) imposes DP on the policy updates, whereas our framework uses Lemma 2 after private reward learning, as post-processing. We return to this distinction in Section 3.

Our analysis focuses on the policy quality under the true reward $r^*$. Let $\pi_{r^*}^\eta \in \arg\max_\pi V_\eta(\pi; r^*)$.

We quantify the $\eta$-regularized suboptimality of a candidate policy $\pi$ by

$$\Delta_\eta(\pi) = V_\eta(\pi_{r^*}^\eta; r^*) - V_\eta(\pi; r^*). \tag{5}$$

In the proposed decoupled framework, $\pi$ will be induced by a private reward estimate, and our theory controls $\Delta_\eta(\pi)$ in finite samples.

## 2.2 Differential Privacy

Informally, a mechanism $M$ is DP if the distribution of its output $M(D)$ is nearly indistinguishable from that of $M(D')$ for any adjacent dataset $D'$, where $D$ and $D'$ differ by a single entry. We formalize the notion of adjacent as their Hamming distance is one.

Among various characterizations of DP–such as Rényi DP (Mironov, 2017) or Gaussian DP (Dong et al., 2022)–aimed at achieving tighter privacy accounting, the following $(\varepsilon, \delta)$-DP remains the most prevalent and standard definition, and thus we also adopt the following:

**Definition 3** $((\varepsilon, \delta)$-Differential Privacy (Dwork et al., 2006)). *A mechanism $M$ is $(\varepsilon, \delta)$-differentially private if, for any two adjacent $D, D'$ and for any measurable event $E$,*

$$\mathbb{P}\big(M(D) \in E\big) \le e^\varepsilon \, \mathbb{P}\big(M(D') \in E\big) + \delta.$$

*When $\delta = 0$, the mechanism $M$ is said to satisfy $\varepsilon$-differential privacy.*

DP protects individuals by requiring that an algorithm produce similar output distributions on adjacent datasets that differ in a single record. This similarity limits the influence of any one data point on what is released, so an observer cannot reliably infer whether that record was included. The $(\varepsilon, \delta)$ definition measures similarity through a log-likelihood ratio control between the two output distributions for every measurable event. The parameter $\varepsilon$ bounds the magnitude of this log-likelihood ratio, while $\delta$ allows a small probability mass on which the bound may be violated, which can be viewed as an $\varepsilon$-DP guarantee holding with probability at least $1 - \delta$. Smaller values of $\varepsilon$ and $\delta$ therefore correspond to stronger privacy protection, and $\delta$ is typically chosen to be extremely small in practice, since it quantifies the probability of a rare failure event the privacy protection may not hold.

A key appeal of DP is that its guarantees are accompanied by explicit parameters that can be tracked across an entire pipeline, which makes it possible to quantify the overall level of protection. In particular, DP enjoys the following properties:

- **Post-processing:** If $M$ is $(\varepsilon, \delta)$-DP, then for any mapping Proc independent of the data, the post-processed mechanism $\text{Proc} \circ M$ is also $(\varepsilon, \delta)$-DP.

- **Sequential composition (basic):** Let $M_1 : \mathcal{X} \to \mathcal{Y}_1$ and $M_i : \mathcal{X} \times \mathcal{Y}_1 \times \cdots \times \mathcal{Y}_{i-1} \to \mathcal{Y}_i$, for $i = 2, \ldots, k$. Let $M(D) := (M_1(D), M_2(D, y_1), \ldots, M_k(D, y_1, \ldots, y_{k-1}))$ denote the joint mechanism, where the outputs are generated recursively. If, for every fixed previous output history, each step is $(\varepsilon_i, \delta_i)$-DP, then $M$ is $\left( \sum_{i=1}^k \varepsilon_i, \ \sum_{i=1}^k \delta_i \right)$-DP.

- **Parallel composition:** Let $D$ be partitioned into disjoint subsets $D_1, \ldots, D_k$, and let $M_i$ be an $(\varepsilon_i, \delta_i)$-DP mechanism applied only to $D_i$. Define the joint mechanism $M(D) := \big(M_1(D_1), \ldots, M_k(D_k)\big)$. Then $M$ is $\Big(\max_{1 \leq i \leq k} \varepsilon_i,\ \max_{1 \leq i \leq k} \delta_i\Big)$-DP.

We present the basic sequential composition rule only for illustration, and the main point is that privacy parameter accumulates across multiple data-dependent releases. In practice, one can obtain tighter accounting by using refined composition theorems and privacy definitions designed for sharper tracking of cumulative loss, such as Rényi DP or Gaussian DP. We do not pursue these refinements here since our focus is on the pipeline-level design and on policy-quality guarantees, rather than on the tightest possible accounting constants.

Constructing a DP mechanism typically involves injecting noise, calibrated to the *sensitivity*, the maximum impact of a single data point on the output. In machine learning where gradient-based optimization is standard, the contribution of an individual data is its respective gradient. Therefore, a natural pathway to DP is to inject noise into the gradient updates, leading to the DP-stochastic gradient descent (DP-SGD) (Abadi et al., 2016).

Concretely, let $\ell(\theta; z)$ denote a per-example loss function and $\Theta \subset \mathbb{R}^d$ be a parameter space. A standard minibatch DP-SGD update is defined as:

$$\theta_{t+1} = \Pi_\Theta \left( \theta_t - \eta_t \left( \frac{1}{|B_t|} \sum_{i \in B_t} \mathrm{clip}(\nabla_\theta \ell(\theta_t; Z_i), C) + \xi_t \right) \right), \quad \xi_t \sim \mathcal{N}(0, \sigma_{\mathrm{DP}}^2 C^2 I_d), \qquad (6)$$

where $B_t \subset [n]$ is a minibatch, $\Pi_\Theta$ denotes the projection onto the set $\Theta$, and $\mathrm{clip}(g, C) := g \cdot \min\{1, C/\|g\|_2\}$ scales each gradient to ensure a uniform $\ell_2$-sensitivity bound $C$. While sensitivity may not be inherently bounded or analytically tractable–particularly for black-box models– this gradient clipping ensures that the sensitivity is always bounded by $C$, which is then used to calibrate the noise. As such, gradient clipping serves as a key tool that provides the flexibility to satisfy DP for arbitrary differentiable architectures.

Other optimization approaches are primarily distinguished by the stage at which noise is introduced, such as objective perturbation or output perturbation (Chaudhuri et al., 2011). For large-scale tasks, however, DP-SGD remains the *de facto* standard and the most widely used scalable paradigm in practice, supported by implementations such as the `Opacus` library in Python (Yousefpour et al., 2021).

## 3 Proposed Method

### 3.1 Motivation: Challenges in Private Policy Optimization

The prevailing paradigms in finetuning, such as DPO and PPO, optimize the policy parameters $\pi_\theta$ directly. We defer a more detailed description of PPO to Appendix B.1, and focus here on the common issue that arises when DP is imposed on policy optimization.

The fundamental difficulty stems from the inherent incompatibility between the unbounded nature of policy gradients and the sensitivity required by DP. In frameworks like PPO or DPO,

the gradient involves the score function

$$\nabla_\theta \log \pi_\theta(a|x) = \frac{\nabla \pi_\theta(a|x)}{\pi_\theta(a|x)},$$

which lacks a uniform upper bound since $\pi_\theta(a|x)$ can be arbitrarily close to zero, making the ratio arbitrarily large. As these policy gradients can be arbitrarily large, the aforementioned clipping during the gradient updates are necessary.

**Example 4** (DPO). *Consider the DPO objective in* (4). *For a single preference pair* $(x, a^w, a^l)$, *define*

$$z_\theta(x, a^w, a^l) := \frac{1}{\eta} \log \frac{\pi_\theta(a^w|x)}{\pi_0(a^w|x)} - \frac{1}{\eta} \log \frac{\pi_\theta(a^l|x)}{\pi_0(a^l|x)}.$$

*The per-pair loss is* $\ell(\theta) = -\log \sigma(z_\theta(x, a^w, a^l))$. *Since* $\pi_0$ *is fixed, the gradient is*

$$\nabla_\theta \ell(\theta) = -(1 - \sigma(z_\theta)) \nabla_\theta z_\theta = -\frac{1}{\eta}(1 - \sigma(z_\theta))\Big(\nabla_\theta \log \pi_\theta(a^w \mid x) - \nabla_\theta \log \pi_\theta(a^l \mid x)\Big).$$

*This expression depends on the score function* $\nabla_\theta \log \pi_\theta(a \mid x)$, *which generally admits no uniform bound because* $\pi_\theta(a \mid x)$ *can be arbitrarily small. As a result, per-example gradients can be arbitrarily large under standard policy parameterizations, and DP training therefore requires explicit sensitivity control through per-example gradient clipping.*

**Example 5** (PPO-style policy optimization). *A similar issue arises for PPO-style policy optimization. At a high level, PPO uses an advantage signal to indicate whether a sampled action should become more likely or less likely under the updated policy, while constraining the update so that the policy does not move too far in a single step. Here the advantage* $A$ *is a scalar quantity summarizing how favorable action* $a$ *is at context* $x$ *relative to a baseline. The PPO-style ratio*

$$\rho_\theta(x, a) := \frac{\pi_\theta(a \mid x)}{\pi_{\text{ref}}(a \mid x)} = \exp\Big(\log \pi_\theta(a \mid x) - \log \pi_{\text{ref}}(a \mid x)\Big)$$

*measures how much the updated policy reweights action* $a$ *relative to a fixed reference policy* $\pi_{\text{ref}}$. *Given a parameter* $\varepsilon_{\text{clip}} > 0$, *the clipped surrogate takes the form*

$$\ell_{\text{PPO}}(\theta) = -\min\Big\{\rho_\theta(x, a)\, A,\ \text{clip}(\rho_\theta(x, a), 1 - \varepsilon_{\text{clip}}, 1 + \varepsilon_{\text{clip}})\, A\Big\}.$$

*Whenever the unclipped branch is active, differentiating with respect to* $\theta$ *yields*

$$\nabla_\theta \rho_\theta(x, a) = \rho_\theta(x, a)\, \nabla_\theta \log \pi_\theta(a \mid x),$$

*so the gradient again depends on the score function* $\nabla_\theta \log \pi_\theta(a \mid x)$. *Under standard policy parameterizations, this score function generally admits no uniform bound because* $\pi_\theta(a \mid x)$ *can be arbitrarily small. Thus, PPO clipping controls the objective through the policy ratio, but it does not remove the need for explicit per-example gradient clipping when DP is imposed.*

Clipping plays two roles in private training. It enforces a sensitivity bound, but it can also distort optimization when many per-example gradients exceed the clipping threshold. Let

$F(\theta) = \mathbb{E}_z[\ell(\theta; z)]$ and define the minibatch gradient $\widehat{\nabla} F_t(\theta)$. Under standard sampling assumptions, $\widehat{\nabla} F_t(\theta)$ is an unbiased estimator of $\nabla F(\theta)$, whereas replacing per-example gradients by $\mathrm{clip}(\nabla_\theta \ell(\theta; z), C)$ generally introduces bias.

This distinction is especially relevant in policy optimization, where per-example gradients can be heavy-tailed due to score-function terms, so clipping may be frequently active. In contrast, when the learning target admits a uniform per-example gradient bound, one can choose $C$ to match this bound so that clipping is rarely active and introduces negligible distortion. In that case, privacy is enforced primarily through additive noise calibrated to the sensitivity bound, which preserves mean-zero updates while inflating variance.

To address these challenges, we place the privacy mechanism on reward learning rather than on policy optimization. The reward-modeling objective is typically better conditioned, and in many LLM pipelines it is implemented by freezing a pretrained backbone and training a lightweight head on top of its representations (Evci et al., 2022; Houlsby et al., 2019; Hu et al., 2022). Let $\phi(x,a) \in \mathbb{R}^d$ denote the fixed representation of a context action pair, and consider a linear-head reward model

$$r_\theta(x,a) = \langle \phi(x,a), \theta \rangle, \tag{7}$$

with $\theta \in \Theta = \{\theta \in \mathbb{R}^d \mid \|\theta\|_2 \le R\}$. The key point is that sensitivity control becomes transparent once the per-example loss is Lipschitz in $\theta$. A convenient way to ensure this is to keep the representation norm bounded, for instance by applying a final normalization step on the backbone features. In modern Transformers, LayerNorm and RMSNorm already stabilize feature scales in practice (Ba et al., 2016; Zhang and Sennrich, 2019; Zheng et al., 2024), and an explicit final projection or normalization can enforce a deterministic bound $\sup_{x,a} \|\phi(x,a)\|_2 \le L$.

Under the Bradley–Terry model, each observation $z = (x, a^w, a^\ell)$ induces a convex loss $\ell(\theta; z)$ whose gradient has the form

$$\nabla_\theta \ell(\theta; z) = \alpha(\theta; z)(\phi(x, a^w) - \phi(x, a^\ell)), \qquad \alpha(\theta; z) \in [0, 1]. \tag{8}$$

Therefore, we have

$$\|\nabla_\theta \ell(\theta; z)\|_2 \le \|\phi(x, a^w) - \phi(x, a^\ell)\|_2 \le \|\phi(x, a^w)\|_2 + \|\phi(x, a^\ell)\|_2 \le 2L. \tag{9}$$

This yields a uniform per-example gradient bound, which provides a clean sensitivity control for private optimization. This suggests that per-example gradients in head-only reward learning are typically better behaved than in policy optimization. As a result, when the same per-example gradient clipping norm $C$ is used to control DP sensitivity, clipping tends to be less frequently active in reward learning than in policy optimization. In that sense, privacy in reward learning is driven more by calibrated noise than by clipping-induced distortion.

## 3.2 Proposed Framework: Private Reward-Based Alignment

We propose a decoupled framework that concentrates the privacy expenditure solely on estimating the reward structure and derives the final decision rule via post-processing.

As outlined in Algorithm 1, the procedure consists of two stages: (i) learning a differen-

---

**Algorithm 1** Differentially Private Reward-Based Alignment

---

1: **Input:** Preference dataset $\mathcal{D} = \{(x_i, a_i^w, a_i^l)\}_{i=1}^n$, privacy budget $(\varepsilon, \delta)$, KL-regularization parameter $\eta$, reference policy $\pi_0$, sampling budget $N$ (optional).
2: **Step 1: Private Reward Learning**
3: Learn a private reward model $\tilde{r}(x, a)$ by an $(\varepsilon, \delta)$-DP mechanism (e.g., DP-SGD).
4: **Step 2: Policy Derivation (Inference)**
5: **if** normalization constant $Z_{\tilde{r}}(x) = \mathbb{E}_{a \sim \pi_0}[\exp(\eta \tilde{r}(x, a))]$ is computable **then**
6:    *Exact Inference*
7:    Construct the optimal policy in closed form:

$$\pi_{\tilde{r}}^\eta(a|x) = \frac{1}{Z_{\tilde{r}}(x)} \pi_0(a|x) \exp\left(\eta \tilde{r}(x, a)\right).$$

8: **else**
9:    *Approximate Inference via Best-of-N*
10:    For a given context $x$, sample $N$ candidates from reference: $\{a^{(1)}, \ldots, a^{(N)}\} \sim \pi_0(\cdot|x)$.
11:    Select the candidate maximizing the private reward:

$$a^* = \underset{j \in \{1, \ldots, N\}}{\operatorname{argmax}} \tilde{r}(x, a^{(j)}).$$

12:    Define the policy output as the Dirac mass on $a^*$.
13: **end if**
14: **Output:** Privately aligned policy $\pi_{\tilde{r}}^\eta$ or action $a^*$.

---

tially private reward model $\tilde{r}$ from the full preference dataset $\mathcal{D}$, and (ii) producing an aligned action/response induced by $\tilde{r}$ without any further access to $\mathcal{D}$.

In the first stage, we treat reward learning as a single empirical risk minimization problem on $\mathcal{D}$. We state the framework at the level of the resulting $(\varepsilon, \delta)$-DP guarantee, rather than specifying an explicit closed-form calibration for the added Gaussian noise. In practice, the calibration of DP-SGD depends on various factors, including the clipping norm, sampling scheme, number of epochs, and privacy accounting method (Abadi et al., 2016; Bu et al., 2020). Since our focus is on the pipeline design and on the privacy–utility trade-off at the level of the final guarantee, rather than on accountant-specific calibration formulas, we do not make the noise level explicit here. In the theoretical development in Section 4, we work with the projected noisy stochastic-gradient procedure of Bassily et al. (2014) as a concrete DP-SGD instantiation, since it delivers the strongly-convex excess-risk rate used in our analysis. In the experiments, the corresponding calibration is handled by `Opacus`, which takes the target privacy parameters together with the training configuration and internally performs privacy accounting to determine the required noise level; implementation details are deferred to Appendix B.4 and Table 4.

In the second stage, the KL-regularized target policy induced by $\tilde{r}$ is defined by the Gibbs form

$$\pi_{\tilde{r}}^\eta(a \mid x) = \frac{\pi_0(a \mid x) \exp\{\eta \tilde{r}(x, a)\}}{Z_{\tilde{r}}^\eta(x)},$$

where $Z_{\tilde{r}}^\eta(x) := \sum_{a' \in \mathcal{A}} \pi_0(a' \mid x) \exp\{\eta \tilde{r}(x, a')\}$. The implementation depends on the tractability of the partition function $Z_{\tilde{r}}^\eta(x)$. When $Z_{\tilde{r}}^\eta(x)$ is tractable (e.g., finite action sets or structured settings where normalization is feasible), one can explicitly construct and sample from $\pi_{\tilde{r}}^\eta(\cdot \mid x)$

(or apply a deterministic rule such as $\arg\max_a \pi_{\tilde{r}}^\eta(a \mid x)$).

In contrast, when $\mathcal{A}$ is combinatorially large–as in preference fine-tuning for LLMs where actions correspond to long token sequences–computing $Z_{\tilde{r}}^\eta(x)$ is infeasible. In this regime, Algorithm 1 adopts a best-of-$N$ (BoN) inference-time policy (Stiennon et al., 2020): draw $N$ candidates $a^{(1)}, \dots, a^{(N)} \sim \pi_0(\cdot \mid x)$ and output

$$a^* \in \arg\max_{j \in \{1,\dots,N\}} \tilde{r}(x, a^{(j)}).$$

As $N$ increases, this selection increasingly tends to return higher-reward candidates under the proposal distribution $\pi_0(\cdot \mid x)$ by restricting attention to a richer candidate pool, without requiring explicit normalization over $\mathcal{A}$.

A simple rationale for using $\pi_0$ as a proposal distribution follows from the relationship between the KL-regularized target policy and the reference. Consider $\pi_{\eta r}(a|x) \propto \pi_0(a|x) \exp\{\eta\, r(x, a)\}$. If the reward is uniformly bounded, i.e., $\sup_{x \in \mathcal{X}, a \in \mathcal{A}} |r(x, a)| \leq B$ for some $B < \infty$, then for every $(x, a)$,

$$e^{-\eta B} \;\leq\; \frac{\pi_{\eta r}(a|x)}{\pi_0(a|x)} \cdot Z_{\eta r}(x) \;\leq\; e^{\eta B},$$

with $Z_{\eta r}(x) := \sum_{a' \in \mathcal{A}} \pi_0(a'|x) \exp\{\eta r(x, a')\}$. Since $e^{-\eta B} \leq Z_{\eta r}(x) \leq e^{\eta B}$, we obtain the pointwise bounds

$$e^{-2\eta B} \;\leq\; \frac{\pi_{\eta r}(a|x)}{\pi_0(a|x)} \;\leq\; e^{2\eta B}.$$

In particular, when $\eta B$ is moderate, $\pi_{\eta r}(\cdot|x)$ remains within a controlled multiplicative tilt of $\pi_0(\cdot|x)$, which supports using $\pi_0(\cdot|x)$ as a reasonable proposal distribution for candidate-based approximate inference.

We highlight that Algorithm 1 is designed to exploit a structural feature specific to RLHF. In standard RL, a separate reward-learning stage is absent because rewards are observed directly from the environment. As a result, if one seeks to enforce DP in standard RL, privacy must be introduced at the policy-optimization stage, typically through noisy policy gradient updates (He and Zhou, 2025). RLHF is different in that it introduces an intermediate reward-learning layer. Our framework places DP on this layer and derives the final policy by post-processing the resulting private reward model.

The role of the reward-learning layer is also useful for comparing our framework with DP-DPO. As discussed in Section 2, DPO uses the identity in (3) as a change of variables from rewards to policies. When the policy class is tied to the reward class through (3), the DPO pairwise likelihood uses the Bradley–Terry reward difference as its pairwise logit. The difference under DP is where the private mechanism enters. DP-DPO applies the privacy mechanism to the policy update, whereas our framework applies it to reward learning and then uses (3) to derive a policy from the private reward model. As illustrated in Examples 4 and 5, policy-level privatization exposes the update to the clipping–noise tradeoff of policy gradients. In our framework, the exact Gibbs policy or the Best-of-$N$ rule is constructed after the private reward model is learned and therefore incurs no additional privacy cost.

In addition, placing DP only on reward learning separates our framework from multi-stage private RLHF pipelines. In approaches that privatize both reward learning and policy optimiza-

tion (Wu et al., 2024, 2025), the second private stage requires either dividing the privacy budget or splitting the data across stages. Our procedure avoids this second private stage by treating policy derivation as post-processing.

Once the private reward model $\tilde{r}$ is learned, any subsequent output construction, whether through the exact policy or the BoN rule, depends on the dataset only through $\tilde{r}$. This leads to the following privacy guarantee for the entire pipeline.

**Proposition 6** (Privacy of the Framework). *Suppose the reward-learning mechanism $\mathcal{M}$ that outputs a reward model $\tilde{r} = \mathcal{M}(\mathcal{D})$ satisfies $(\varepsilon, \delta)$-DP. Then Algorithm 1 also $(\varepsilon, \delta)$-DP.*

*Proof.* This follows directly from the post-processing property of DP: composing an $(\varepsilon, \delta)$-DP mechanism with any data-independent mapping (including additional randomness independent of $\mathcal{D}$) preserves the same $(\varepsilon, \delta)$-DP guarantee. $\qquad\square$

## 4 Theoretical Analysis

To present a series of theoretical results, we begin by providing essential assumptions.

**Assumption 1** (i.i.d. preference data). *The contexts $x_1, \ldots, x_n$ are i.i.d. draws from $d_0$. For each $i$, two candidate actions are drawn independently from the reference policy $\pi_0(\cdot \mid x_i)$, and $(a_i^w, a_i^l)$ is obtained according to the Bradley–Terry model in Definition 1.*

**Assumption 2** (Linear reward realizability). *There exists a parameter $\theta^* \in \Theta \subset \mathbb{R}^d$, where $\Theta$ is a compact parameter space, such that, for all $(x, a) \in \mathcal{X} \times \mathcal{A}$, $r^*(x, a) = r_{\theta^*}(x, a) = \langle \phi(x, a), \theta^* \rangle$. Moreover, the representation is uniformly bounded: $\sup_{x \in \mathcal{X}, a \in \mathcal{A}} \|\phi(x, a)\|_2 \leq L$.*

**Assumption 3** (Non-degeneracy feature). *Define $\Delta\phi(x; a, a') := \phi(x, a) - \phi(x, a')$. Then the smallest eigenvalue of the matrix $\mathbb{E}_{x \sim d_0, \ a, a' \sim \pi_0(\cdot|x)}\big[\Delta\phi(x; a, a')\,\Delta\phi(x; a, a')^\top\big]$ is $\lambda > 0$.*

**Assumption 4** (Coverage). *There exists a constant $C$ such that for any $\pi \in \Pi$,*

$$\max_{x, a: d_0(x) > 0} \frac{\pi(a|x)}{\pi_0(a|x)} \leq C,$$

*with convention that $\frac{0}{0} = 0$.*

Assumption 1 specifies the basic offline data-collection model used in our analysis. It provides the independence structure needed for the statistical arguments and is consistent with a common RLHF pipeline in which prompts are sampled, candidate responses are generated from a fixed reference policy, and human pairwise preferences are then collected (Ouyang et al., 2022; Bai et al., 2022).

Assumption 2 specifies a fixed representation $\phi(x, a)$ and assumes that the reward is represented by a linear scoring rule on top of this representation. In LLM applications, $\phi(x, a)$ may be the representation produced by a pretrained backbone before the final reward head, and can encode nonlinear relationships between the prompt $x$ and response $a$. Thus, the linearity assumption applies only to the final reward-scoring layer, not to the representation map itself. This type of fixed-representation reward realizability assumption is standard in theoretical analyses of preference-based policy learning and RLHF (Zhu et al., 2023; Liu et al., 2025b). The

boundedness condition on $\phi(x, a)$ is a regularity requirement that ensures the reward class is well behaved and supports clean sensitivity and concentration arguments. This representation-head formulation also aligns with common lightweight adaptation practice in LLMs, where a pretrained backbone is used as a representation map and only a small downstream component is trained for the target objective (Houlsby et al., 2019; Zaken et al., 2022; Hu et al., 2022; Evci et al., 2022). Under this head-only design, linear reward modeling connects the practical frozen-backbone pipeline to a tractable theoretical analysis.

Assumption 3 provides the non-degeneracy needed for identifiability and strong convexity in the pairwise-difference parameterization. The Bradley–Terry likelihood depends on $\theta$ only through $\langle \theta, \Delta\phi(x; a, a') \rangle$. Hence any direction $v$ satisfying $v^\top \Delta\phi(x; a, a') = 0$ almost surely is not identifiable from pairwise comparisons. The positive-definiteness of $\mathbb{E}[\Delta\phi(x; a, a')\Delta\phi(x; a, a')^\top]$ rules out such null directions. Under Assumptions 2 and 3, this non-degeneracy also yields high-probability strong convexity on the identified pairwise-difference parameterization. Similar non-degeneracy conditions are standard in theoretical analyses (Zhu et al., 2023; Zhong et al., 2024; Liu et al., 2025b).

Assumption 4 imposes a uniform bound on the density ratio between candidate policies and the reference policy $\pi_0$. It rules out policies that place substantial mass on actions that have negligible probability under the data-generating policy $\pi_0$. Under Assumption 1, candidate responses are generated from $\pi_0$, so the preference data provide essentially no information about responses that $\pi_0$ does not generate. Coverage assumptions formalize this limitation and are standard in RLHF and related RL theory (Munos and Szepesvári, 2008; Zhan et al., 2022; Uehara and Sun, 2021; Xiong et al., 2024; Song et al., 2024). In LLM preference fine-tuning, this means that the guarantees are intended for policies whose responses remain supported by the SFT or instruction-tuned reference model, rather than for response regions assigned negligible probability by the reference model (Ouyang et al., 2022; Rafailov et al., 2024).

## 4.1 Upper Bound on the Suboptimality Gap

We first deliver the utility analysis of our private estimator. The following lemma, adapted from Bassily et al. (2014), characterizes the expected excess empirical risk.

**Lemma 7** (Utility of private projected SGD). *Suppose Assumption 1, 2 and Assumption 3 hold. Let $\tilde{\theta}_n$ be the output of DP-SGD procedure of Bassily et al. (2014); that is, at each iteration, one data point is sampled uniformly with replacement, Gaussian noise is added to the resulting stochastic gradient, and the update is projected back onto $\Theta$, where the Gaussian noise is calibrated to satisfy $(\varepsilon, \delta)$-DP under the per-example gradient bound $2L$. Fix any $\rho \in (0, 1)$ and let $n \geq \frac{32L^2}{\lambda}\log\left(\frac{d}{\rho}\right)$. Then there exists an event $\mathcal{E}$ with $\mathbb{P}(\mathcal{E}) \geq 1 - \rho$ such that, on $\mathcal{E}$, the negative log-likelihood is $\mu$-strongly convex over $\Theta$ with $\mu = \frac{\lambda}{2}\sigma(2RL)(1 - \sigma(2RL))$, where $\sigma(t) = (1 + e^{-t})^{-1}$ is the sigmoid function. Moreover, on the same event $\mathcal{E}$,*

$$\mathbb{E}\left[\bar{L}_n(\tilde{\theta}_n) - \bar{L}_n(\hat{\theta}_n) \,\big|\, D\right] = \tilde{O}\left(\frac{d}{n^2\varepsilon^2}\right),$$

*where the expectation is over the algorithmic randomness conditional on $D$.*

**Proof sketch.** The Bradley–Terry likelihood depends on $\theta$ only through pairwise reward differences, so strong convexity is considered on the identified pairwise-difference parameterization. Assumption 3, together with bounded features and bounded $\Theta$, gives a positive population curvature on this parameterization. A matrix concentration bound then shows that the empirical loss inherits this strong convexity on an event $\mathcal{E}$ with $\mathbb{P}(\mathcal{E}) \geq 1 - \rho$. Conditioning on $\mathcal{E}$, the per-example loss is $2L$-Lipschitz and strongly convex, so the standard strongly-convex DP-SGD utility guarantee yields the stated conditional expected excess empirical risk bound.  $\square$

Lemma 7 provides a utility bound for our private reward estimator by certifying, with high probability, that the empirical objective is strongly convex over $\Theta$. Various variants of DP-SGD are available, and for the privacy guarantee itself our framework is compatible with any such instantiation that attains the target $(\varepsilon, \delta)$-DP. For the theoretical analysis, however, we invoke the strongly-convex private stochastic-gradient guarantee of Bassily et al. (2014), which yields the optimal excess-risk order in the strongly-convex regime. In our linear-head setting, the per-example gradient is uniformly bounded by $2L$, so the same type of guarantee applies after calibrating the noise to the target $(\varepsilon, \delta)$-DP. This allows us to quantify the privacy-induced optimization error at the sharpest available rate without postulating strong convexity as a separate assumption, and we therefore state the lemma in terms of the resulting privacy guarantee and excess empirical risk rate rather than in terms of an accountant-specific calibration formula.

**Remark 8** (Unconditional DP-SGD control). *To characterize the suboptimality gap, we require a high-probability control of the estimation error with respect to the joint randomness of the data and the learning procedure. Much of the existing DP-SGD literature analyzes the procedure under a fixed dataset (or in expectation), which does not directly provide the unconditional statement needed to our analysis. Section D.2 in the supplementary provides the missing bridge by proving an unconditional result tailored to our setup (Theorem 11).*

We now state our main upper bound on the suboptimality gap.

**Theorem 9** (Upper bound on the suboptimality gap). *Suppose Assumptions 1, 2, 3, and 4 hold. Consider Algorithm 1 instantiated with the private projected SGD procedure in Lemma 7, and let $\tilde{\theta}$ be its output. Let $\pi_{\tilde{\theta}}^{\eta}$ denote the induced KL-regularized policy. Fix any $\rho \in (0, 1)$. If $n \geq \frac{32L^2}{\lambda} \log\left(\frac{d}{\rho}\right)$, then with probability at least $1 - \rho$,*

$$\Delta_{\eta}\left(\pi_{\tilde{\theta}}^{\eta}\right) \leq \tilde{O}\left(\frac{\eta d}{n} + \frac{\eta d}{n^2 \varepsilon^2}\right).$$

Theorem 9 provides an additive decomposition of the suboptimality gap into a non-private term of order $\eta d/n$ and a privacy cost of order $\eta d/(n^2 \varepsilon^2)$. The parameter $\delta$ which characterizes a failure probability of the privacy protection is typically chosen to be negligible, e.g., $\delta = n^{-k}$ for some $k \geq 2$; under such choices, its impact enters only through logarithmic factors and does not affect the leading rates. Notably, privacy does not worsen the dimensional dependence, so the price of privacy appears only through the extra $1/(n\varepsilon^2)$ factor.

To interpret the sample-size regimes, define the crossover scale $n_{\varepsilon} := \varepsilon^{-2}$. When $n \gtrsim n_{\varepsilon}$, the privacy-induced term is lower order and the rate effectively matches the non-private one; when $n \lesssim n_{\varepsilon}$, the privacy cost can dominate and determines the attainable gap. This captures

the privacy–utility tradeoff: holding $\varepsilon$ fixed recovers the non-private rate as $n$ grows, while strengthening privacy by shrinking $\varepsilon$ increases the privacy-induced term. Importantly, the faster decay $1/n^2$ in the privacy term arises from invoking strongly-convex DP optimization guarantees on a high-probability event; without such curvature, privacy costs would typically decay only as $1/n$ rather than $1/n^2$.

The bound is linear in the KL regularization parameter $\eta$, which we treat as fixed throughout the analysis, as is standard in KL-regularized RLHF and related formulations (Ouyang et al., 2022; Rafailov et al., 2024; Zhao et al., 2024). This linear dependence is natural, since smaller $\eta$ pulls both $\pi_\theta^\eta$ and $\pi_{\theta*}^\eta$ closer to the reference policy $\pi_0$, thereby reducing the gap between the two induced policies.

## 4.2   Minimax Lower Bound on the Suboptimality Gap

We next establish a minimax lower bound on the suboptimality gap for the $d$-dimensional linear reward model class, which applies to any $(\varepsilon, \delta)$-DP algorithm.

**Theorem 10** (Minimax lower bound)**.** *Fix $\eta > 0$ and consider the $d$-dimensional linear reward model class. For $\varepsilon \in (0, 1]$ and $\delta \leq \varepsilon$, define the minimax risk*

$$R_n(\varepsilon, \delta) := \inf_{A \in \mathcal{A}_{\varepsilon,\delta}} \sup_{\theta^* \in \Theta} \mathbb{E}_{\theta^*}[\Delta_\eta(A(Z^n))],$$

*where $\mathcal{A}_{\varepsilon,\delta}$ is the class of $(\varepsilon, \delta)$-DP algorithms and $Z^n$ denotes $n$ preference pairs generated under $\theta^*$.*

*Then, for each fixed $\eta > 0$, there exist constants $c_\eta, C_\eta > 0$ such that, up to logarithmic factors, for all $n \geq n_{\mathrm{NP}} := C_\eta d$,*

$$R_n(\varepsilon, \delta) \geq c_\eta \max\left\{\frac{d}{n}, \ \min\left(\frac{1}{n\varepsilon}, \ \frac{d}{n^2\varepsilon^2}\right)\right\}.$$

*Moreover, letting $n_{\mathrm{P}} := C_\eta d/\varepsilon$, for all $n \geq \max\{n_{\mathrm{NP}}, n_{\mathrm{P}}\}$,*

$$R_n(\varepsilon, \delta) \geq c_\eta \max\left\{\frac{d}{n}, \ \frac{d}{n^2\varepsilon^2}\right\}.$$

**Proof sketch.** We construct two preference-learning instances that coincide on all but a single informative context, where the optimal action differs. Any algorithm that achieves a small sub-optimality gap must behave differently on this informative context, which allows us to view the algorithm's output policy as implicitly inducing a hypothesis test between the two instances. DP then creates an additional bottleneck. Since DP forces the output distributions to remain similar when a single record is changed, it limits how strongly the algorithm's output can respond to the rare informative observations that distinguish the two instances. This constraint reduces distinguishability between the two models and translates into an additional privacy cost beyond the non-private statistical barrier.                              $\square$

Theorem 10 gives an information-theoretic limitation that holds uniformly over all $(\varepsilon, \delta)$-DP algorithms in the $d$-dimensional linear reward class. Unlike the upper bound in Theorem 9, where $\eta$ is kept explicit because it directly controls how conservatively the induced policy departs from

$\pi_0$, the minimax lower bound in Theorem 10 treats $\eta$ as fixed and suppresses its dependence in the stated rate. Our main goal here is to isolate the phase transition in $(n, d, \varepsilon)$, namely the non-private term $d/n$, the privacy-dominated term $d/(n^2\varepsilon^2)$, and the pre-asymptotic branch $1/(n\varepsilon)$. While a more refined proof-level expression can retain $\eta$, doing so does not change this regime structure and would only make the theorem statement heavier. For this reason, we keep $\eta$ explicit in the upper bound, where it aids interpretation, but suppress it in the lower bound, where the main message is the $(n, d, \varepsilon)$-dependent scaling.

The first term, $d/n$, is the nonprivate minimax lower bound established in Zhao et al. (2024). Since the class of $(\varepsilon, \delta)$-DP algorithms is a subset of all algorithms considered there, this $\Omega(d/n)$ term necessarily persists under privacy. The remaining terms quantify the additional loss due to privacy.

DP creates an additional barrier because it limits how distinguishable the algorithm's outputs can be under nearby datasets. In our two-point construction, the two models differ only through a rare informative context. Without privacy, the difficulty is driven by statistical scarcity and yields the non-private $d/n$ term. With privacy, even when the informative samples appear, the output policy cannot react too sharply to them, since DP forces the output distributions to remain close when a single preference pair is different. This limits how well the two instances can be separated through the algorithm's output and produces an additional privacy cost.

The privacy-dependent term splits into two regimes. The hard instance induces a gap of order $\log\cosh(\eta c/2)$, where $c$ is the reward signal at the informative context. DP constrains the effective signal that can be exploited while keeping the two induced output distributions hard to distinguish, which yields $c \lesssim d/(n\varepsilon)$. When $n$ is large enough that $\eta c$ lies in the local quadratic region, $\log\cosh(u)$ behaves like $u^2$, and the resulting contribution scales as $d/(n^2\varepsilon^2)$. When $n$ is smaller and $\eta c$ falls outside this region, $\log\cosh(u)$ is closer to linear, which leads to the weaker $1/(n\varepsilon)$ rate.

Finally, we assume $\varepsilon \in (0, 1]$ and $\delta \leq \varepsilon$, which is exactly the condition used when applying Lemma 19, the DP Le Cam testing bound of Acharya et al. (2021), to keep the privacy-dependent testing error bounded away from zero. This condition is also natural from the privacy perspective, since $\delta$ represents the probability of a rare failure event in the privacy guarantee in the $(\varepsilon, \delta)$-DP and is typically chosen to be very small. In particular, standard choices such as $\delta = n^{-k}$ with $k \geq 2$ satisfy $\delta \leq \varepsilon$ for all sufficiently large $n$ when $\varepsilon$ is fixed. A comparison with Theorem 9 then shows that our upper bound matches this lower bound up to logarithmic factors in the regime identified below. We formalize this comparison in the next subsection.

### 4.3 Rate-optimality

We call an algorithm *rate-optimal* if its suboptimality gap matches the minimax risk up to logarithmic factors as a function of $(n, d, \varepsilon)$. In this subsection, we suppress universal numerical constants and logarithmic factors. We denote the privacy scale by $n_\varepsilon = \varepsilon^{-2}$.

Theorem 9 yields the upper bound

$$\Delta_\eta\left(\pi_{\hat{\theta}}^\eta\right) \leq \tilde{O}\left(\frac{d}{n} + \frac{d}{n^2\varepsilon^2}\right).$$

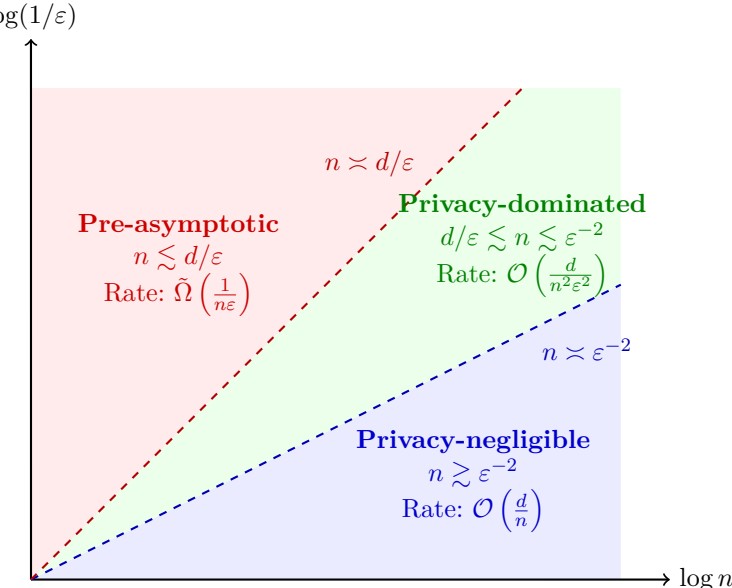

Figure 3: Phase diagram of the statistical and privacy errors on a log-log scale. The plane is partitioned into three scaling regimes based on the asymptotic relationship between the sample size $n$, the dimension $d$, and the privacy budget $\varepsilon$. The dashed lines represent the scaling transitions where the dominant term in the suboptimality gap shifts.

Theorem 10 yields the minimax lower bound

$$R_n(\varepsilon, \delta) \geq \widetilde{\Omega}\left(\max\left\{\frac{d}{n}, \min\left(\frac{1}{n\varepsilon}, \frac{d}{n^2\varepsilon^2}\right)\right\}\right).$$

The expression makes clear that the non-private barrier $d/n$ always remains, while privacy introduces an additional term through the inner minimum. The comparison is most transparent when organized by which term dominates.

- **Privacy-negligible regime ($n \gtrsim n_\varepsilon$).** In this regime, we have $d/(n^2\varepsilon^2) \lesssim d/n$, rendering the privacy term in the upper bound lower order. Since the lower bound always contains $d/n$, both bounds are governed by the non-private rate $d/n$. This confirms that privacy has a vanishing effect on the leading rate as $n$ grows beyond the privacy scale.

- **Privacy-dominated and rate-optimal regime ($d/\varepsilon \lesssim n \lesssim n_\varepsilon$).** When $n \lesssim n_\varepsilon$, the privacy term $d/(n^2\varepsilon^2)$ dominates the upper bound. In this range, the lower bound depends on the inner minimum. Specifically, when $n$ is large enough such that

$$\frac{d}{n^2\varepsilon^2} \leq \frac{1}{n\varepsilon} \iff n \geq \frac{d}{\varepsilon},$$

the minimum selects $d/(n^2\varepsilon^2)$, and the lower bound simplifies to

$$R_n(\varepsilon, \delta) \geq \widetilde{\Omega}\left(\max\left\{\frac{d}{n}, \frac{d}{n^2\varepsilon^2}\right\}\right).$$

Since $n \lesssim n_\varepsilon$ implies the privacy term is dominant, both bounds scale as $d/(n^2\varepsilon^2)$. Thus, the algorithm is rate-optimal in this privacy-dominated regime.

- **Pre-asymptotic regime ($n \lesssim d/\varepsilon$).** When $n$ is small enough that

$$\frac{1}{n\varepsilon} \leq \frac{d}{n^2\varepsilon^2} \iff n \leq \frac{d}{\varepsilon},$$

the inner minimum in the lower bound becomes $1/(n\varepsilon)$. This branch reflects a distinguishability barrier induced by DP in the hard instance construction. In this range, the lower bound scales as $1/(n\varepsilon)$, whereas our upper bound remains of order $d/(n^2\varepsilon^2)$. We do not claim tightness of the upper bound in this regime.

Taken together, the bounds exhibit a transition at the privacy scale $n_\varepsilon$. Above $n_\varepsilon$, the leading rate is governed by statistical error $(d/n)$. Below $n_\varepsilon$, privacy dominates, and the optimal decay becomes quadratic $(d/(n^2\varepsilon^2))$ as long as the sample size is sufficient to enter the local regime $(n \gtrsim d/\varepsilon)$.

## 5   Numerical Studies

In this section, we evaluate our framework empirically. We first use controlled synthetic experiments to validate the theoretical results developed in Section 4 and to compare against private alignment baselines. We then study an LLM fine-tuning experiment to assess practical performance, while implementation details are deferred to Appendix B. Additional numerical results are reported in Appendix C, including reference-underperformance diagnostics in Appendix C.1, sensitivity to the DP-SGD clipping norm in Appendix C.2, and further scaling results with the feature dimension in Appendix C.3.

### 5.1   Synthetic Data Analysis

For the data generation, we define the context dimension $p = \lceil d/2 \rceil$ for each feature dimension $d \in \{3, 5, 7, 9\}$ and sample contexts independently from $x \sim \text{Unif}([-1,1]^p)$. We construct the feature map $\phi(x, a)$ using an interleaved structure of linear terms $x_j$ and centered quadratic terms $q_j(x) = x_j^2 - 1/3$. Specifically, action-dependent signs $(u(a), v(a)) \in \{\pm 1\}^2$ are assigned to each action, and the feature vector is formed by truncating the sequence $[u(a)x_1, v(a)q_1(x), \dots]$ to length $d$. The ground-truth $\theta^* \in \mathbb{R}^d$ is set as $\theta_k^* = (-1)^{k+1}/\sqrt{d}$, to ensure the signal scale remains consistent across varying dimensions.

Regarding the privacy mechanism, we implement the DP-SGD algorithm via the `Opacus` library (Yousefpour et al., 2021). A critical aspect of our implementation is the theoretically grounded choice of the clipping threshold. Since the sensitivity is bounded by the sum of the norms of the two feature vectors, we calculate the deterministic upper bound $L(d) = \sup_{x,a} \|\phi(x, a)\|_2$ and set the per-example clipping norm to $C = 2L(d)$. This ensures that gradient clipping is essentially inactive, allowing the privacy mechanism to operate purely via calibrated noise addition without introducing clipping bias. Unless otherwise stated, we set $\delta = 10^{-5}$. All results are averaged over 30 independent trials.

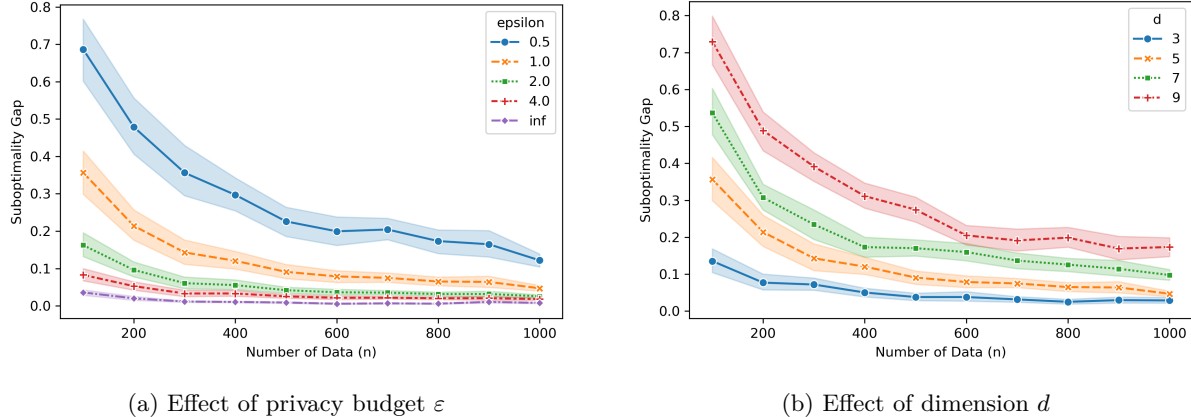

(a) Effect of privacy budget $\varepsilon$  (b) Effect of dimension $d$

Figure 4: **Convergence of Suboptimality Gap.** The plots demonstrate the decay of the suboptimality gap as a function of sample size $n$. (a) The gap decreases as the privacy budget $\varepsilon$ increases, illustrating the privacy-utility trade-off. (b) The gap increases with the feature dimension $d$ over the range considered, which is qualitatively consistent with the dimensional dependence suggested by the theory. Shaded regions indicate 95% confidence intervals over 30 trials.

### 5.1.1 Validation of Theoretical Results

We now investigate the convergence of the suboptimality gap with respect to the sample size $n$, with the goal of validating the theoretical result in Theorem 9. Figure 4 presents the resulting trends of the KL-regularized suboptimality gap.

Figure 4(a) varies the privacy budget $\varepsilon$ while holding $d = 5$ fixed. As $\varepsilon$ increases, the gap decreases across the full range of sample sizes, which is consistent with the privacy-dependent term in the theory. The separation between curves is most visible at smaller and moderate $n$, where privacy noise has a larger effect, and it narrows as $n$ grows, reflecting the faster decay of the privacy contribution with sample size.

Figure 4(b) varies the feature dimension $d$ while holding $\varepsilon = 1.0$ fixed. Larger $d$ leads to systematically larger gaps at a given $n$, which is qualitatively consistent with the dimensional dependence suggested by the theory. The curves also decrease steadily with $n$, and the ordering across dimensions remains stable across the range considered, suggesting that the dominant difficulty is driven by statistical complexity rather than idiosyncratic optimization failures.

### 5.1.2 Comparison to Private Alignment Baselines

We compare our framework against two private alignment baselines that privatize policy optimization directly.

**DP-DPO** applies DP-SGD directly to the DPO objective, optimizing policy parameters under privacy constraints. **DP-RLHF (DP-RM + DP-PPO-like)** splits the preference dataset into two disjoint halves. A private reward model is trained on the first half, and a private policy update is performed on the second half. Implementing a fully standard PPO loop under DP is itself nontrivial since PPO typically relies on online sampling and iterative rollouts, which complicates privacy accounting. Existing work therefore adopts modified PPO-style procedures designed to make privacy accounting tractable (Wu et al., 2024). Following this perspective, we use an *offline pairwise PPO-like* update that avoids actor–critic training and rollout collection.

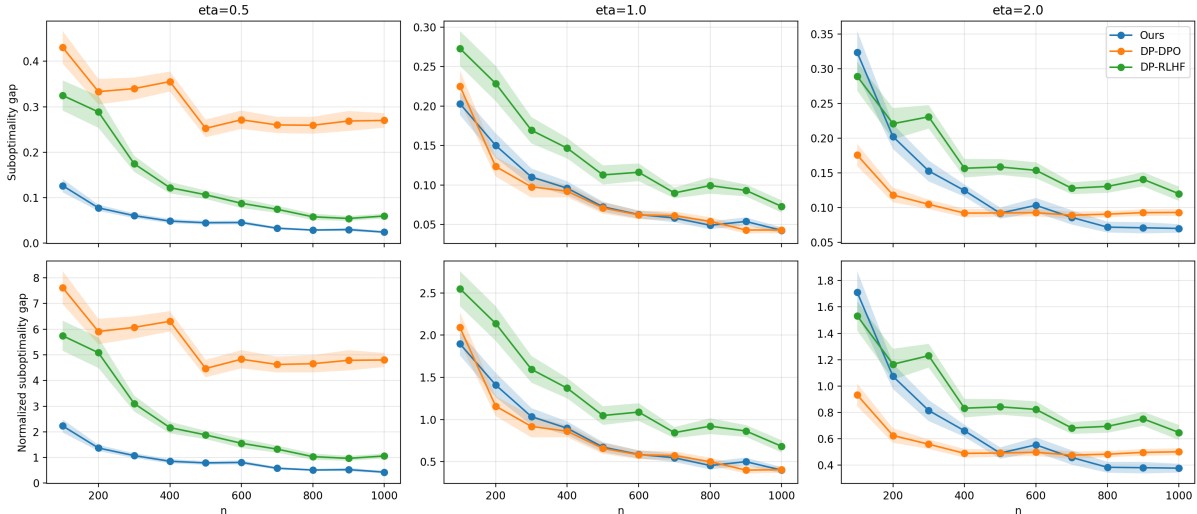

Figure 5: **Synthetic $\eta$-sweep at $(\varepsilon, \delta) = (1, 10^{-5})$ (fixed $d = 7$).** Top row: suboptimality gap $V_\eta(\pi_\eta^\star) - V_\eta(\hat\pi)$. Bottom row: normalized gap. Baselines use $C = 2L(d)$. Shaded regions indicate 95% confidence intervals over 30 trials.

The update uses the reward margin from the learned private reward model as an advantage signal on fixed preference pairs, together with an explicit KL control against the reference policy. By disjointness, the two private stages can each use the full $(\varepsilon, \delta)$ budget via parallel composition. Further implementation details are deferred to Appendix B.1.

We report the *suboptimality gap* (top row in each figure) $V_\eta(\pi_\eta^\star) - V_\eta(\hat\pi)$, and the corresponding *normalized gap* (bottom row)

$$\frac{V_\eta(\pi_\eta^\star) - V_\eta(\hat\pi)}{V_\eta(\pi_\eta^\star) - V_\eta(\pi_0)},$$

where $V_\eta(\pi) = \mathbb{E}[r^*(x,a)] - (1/\eta)\mathrm{KL}(\pi(\cdot \mid x) \,\|\, \pi_0(\cdot \mid x))$. The normalized gap facilitates comparisons across $\eta$ by scaling by the maximum achievable improvement over $\pi_0$. Throughout this section, policy-quality metrics are evaluated on a shared Monte Carlo context set ($N_{\mathrm{eval}} = 2000$).

Figure 5 fixes $(\varepsilon, \delta) = (1, 10^{-5})$ and varies $\eta \in \{0.5, 1, 2\}$. In the conservative regime $\eta = 0.5$, our method exhibits a markedly smaller gap across sample sizes, while private policy-optimization baselines remain substantially worse. For instance, at $n = 1000$, the mean suboptimality gaps are approximately 0.024 (ours), 0.059 (DP-RLHF), and 0.270 (DP-DPO), with the same ordering reflected in the normalized gaps (0.43 vs. 1.06 vs. 4.80). This pattern aligns with the design premise of the paper that when the KL regularization is strong (small $\eta$), injecting DP noise into policy optimization can lead to a poor privacy–utility trade-off, whereas concentrating privacy on reward learning yields a stable guarantee.

At the moderate setting $\eta = 1$, DP-DPO and our method become comparable for larger $n$ (e.g., at $n = 1000$, both attain a gap around 0.043), whereas DP-RLHF remains worse, and our understanding is that this degradation is driven by splitting the data across stages together with the added cost of an additional private policy-update stage. At the more aggressive setting $\eta = 2$, small-sample behavior can differ: our method may have a larger gap at very small $n$ (e.g., at $n = 100$, 0.323 for ours vs. 0.176 for DP-DPO), reflecting amplification of reward-estimation error under a larger $\eta$. However, as $n$ grows, our gap decreases and becomes competitive or

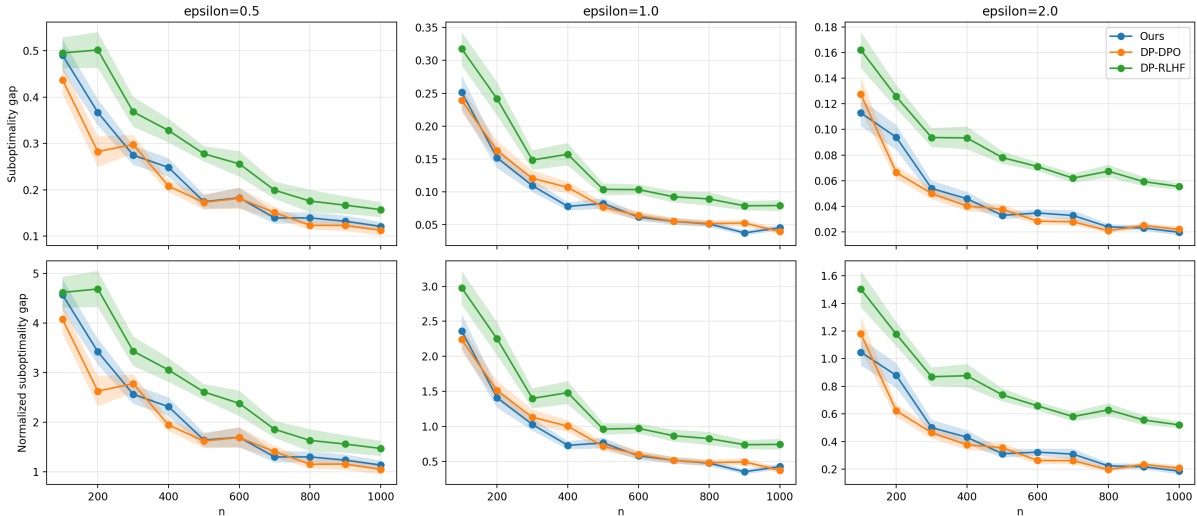

Figure 6: **Synthetic $\varepsilon$-sweep at $\eta = 1$ (fixed $d = 7$).** Top row: suboptimality gap; bottom row: normalized gap. Baselines use $C = 2L(d)$. Shaded regions indicate 95% confidence intervals over 30 trials.

better (e.g., at $n = 1000$, 0.070 for ours vs. 0.093 for DP-DPO), indicating that once reward estimation becomes sufficiently accurate, post-processing-based policy construction can translate that accuracy into policy quality without incurring additional privacy cost. Taken together, the $\eta$-sweep suggests that concentrating privacy on reward learning yields a policy-quality advantage that is relatively robust across regularization levels, which is desirable in settings where $\eta$ is treated as an externally specified departure budget rather than a freely tuned parameter.

Figure 6 fixes $\eta = 1$ and varies $\varepsilon \in \{0.5, 1, 2\}$. As expected, relaxing privacy (larger $\varepsilon$) improves all methods by reducing DP noise in the private updates. Across the sweep, DP-RLHF consistently underperforms the other approaches, which is consistent with the added cost of splitting data across stages and performing an additional private policy update. In contrast, our method and DP-DPO are broadly comparable throughout this $\varepsilon$-sweep. For example, at $n = 1000$ and $\varepsilon = 2$, the mean gap is about 0.020 for our method and 0.022 for DP-DPO, while DP-RLHF remains substantially larger at about 0.055. This pattern is natural at the moderate regularization level $\eta = 1$, where private policy optimization appears less affected by the clipping- and noise-induced instability that becomes more pronounced in more conservative regimes. The clearer advantage of our method emerges in the $\eta$-sweep in Figure 5, particularly for smaller $\eta$, and is further supported by the reference-underperformance diagnostics in Appendix C.1 and the clipping-sensitivity analysis in Appendix C.2. Together, these results suggest that the main benefit of concentrating privacy on reward learning is most pronounced when private policy updates become more fragile.

## 5.2 Application to LLM Finetuning

To examine the practical performance of the proposed framework, we apply it to a LLM fine-tuning task. Across all methods, we use the same reference policy $\pi_0$, `google/gemma-2b-it` (Team et al., 2024), and the same preference dataset, Anthropic HH-RLHF (Bai et al., 2022). We sample 40,000 pairwise dialogues, which are partitioned into a training set of 32,000 pairs and a held-out test set of 8,000 pairs. Although HH-RLHF is curated for helpfulness and harmlessness,

its prompts can still resemble real-world user interactions and may contain health-related, legal, financial, or otherwise personally sensitive context. This makes tuple-level privacy natural in this setting. For additional discussion of the dataset characteristics and privacy motivation, see Appendix B.2. We also match the target privacy budget $(\varepsilon, \delta)$ across methods, while detailed architectural choices, privacy accounting, and hyperparameters are deferred to Appendix B.

For concreteness, we provide a short *paraphrased* example illustrating the structure of a preference tuple:

> **Prompt** ($x$)**:** *Human: I have been struggling with sleep lately. Assistant:*
> **Chosen** ($a^w$)**:** *I am not a clinician, but general steps include sleep hygiene (regular schedule, limiting caffeine) and consulting a professional if symptoms persist.*
> **Rejected** ($a^l$)**:** *Take a prescription sedative; it works for everyone.*

Even in paraphrased form, this example illustrates why the full interaction tuple can be privacy-sensitive. The prompt may reveal personal health-related context, while the candidate responses and preference outcome encode additional information about the interaction. Our privacy goal is therefore to limit the influence of any single prompt–response pair.

For our framework, we instantiate the private reward-learning stage in a head-only form that matches the theoretical setup and linear-probing practice. We freeze the pretrained backbone and train a linear reward head on the final hidden representation. This gives transparent sensitivity control for private reward learning, while also imposing a natural performance ceiling because the backbone is not adapted.

Under the same reward-head architecture, we include two additional reward-based baselines. The label-DP baseline applies randomized response (Warner, 1965; Ghazi et al., 2021) to the binary preference label, keeping the chosen/rejected ordering with probability $e^\varepsilon/(e^\varepsilon + 1)$ and swapping it otherwise. This gives $\varepsilon$-DP for the preference label, but it does not provide tuple-level protection for the full interaction record; we include it as a canonical label-perturbation baseline rather than as a direct privacy-equivalent alternative. The output-perturbation baseline clips scalar reward scores to $[-B, B]$, adds Gaussian noise calibrated to the resulting finite sensitivity, and compares the noisy chosen and rejected scores, giving tuple-level $(\varepsilon, \delta)$-DP for the scalar reward-score release (Chaudhuri et al., 2011).

To align the policy-optimization baselines with the frozen-backbone setup used by our reward-model methods, we avoid full-model private fine-tuning and restrict private policy updates to the final transformer block. All pretrained backbone parameters are frozen, and this final-block update is implemented with LoRA adapters (Hu et al., 2022). DP-RLHF further splits the training data into disjoint reward-modeling and policy-optimization halves.

We highlight that the trainable parameter counts differ because the methods train different terminal components, with a reward head in our framework and LoRA policy adapters in the policy-optimization baselines. The reward-head approaches train 2,048 parameters, whereas the final-block LoRA policy baselines train 102,400 parameters. Nevertheless, both implementations follow a common high-level design principle, under which the pretrained backbone is kept fixed and private training is restricted to the method-specific terminal component. Thus, the comparison fixes the backbone, reference model, and privacy target, while allowing the trained terminal

Table 1: Held-out preference win rate on the HH-RLHF test set. For tuple-DP rows, $\delta = 10^{-5}$; Label-DP is $\varepsilon$-label-DP. Results are averaged over three independent seeds, with standard deviations in parentheses.

| Method | Privacy target | Privacy budget ($\varepsilon$) | | |
| --- | --- | --- | --- | --- |
| | | 0.5 | 1.0 | 2.0 |
| Non-private | None | | 60.33 (0.81) | |
| DP-DPO | Tuple | 51.86 (1.05) | 51.92 (1.12) | 51.99 (1.20) |
| DP-RLHF | Tuple | 53.02 (1.21) | 52.85 (1.59) | 52.82 (1.66) |
| Label-DP | Label | 52.21 (1.45) | 55.89 (0.38) | 59.32 (0.97) |
| Output perturbation | Tuple | 50.46 (0.17) | 50.90 (0.30) | 51.50 (0.44) |
| **Ours: Private RM** | Tuple | **58.93** (0.51) | **59.44** (0.71) | **59.69** (0.62) |

component to differ across methods. Additional details are reported in Appendix B.3.

Evaluation is based on the chosen/rejected pairs in the HH-RLHF test set. For each test pair $(x, y^w, y^l)$, we check whether the learned object ranks the chosen response $y^w$ above the rejected response $y^l$. For policy-optimization baselines, including DP-DPO and DP-RLHF, this ranking is computed from response-only total log-likelihoods under the learned policy. For reward-model methods, including ours, label-DP, and output perturbation, this ranking is computed from the learned reward scores. The reward-based comparison is directly related to the policy induced by our framework. If $\hat{r}$ is the learned reward model, then Lemma 2 gives

$$\log \frac{\pi_{\hat{r}}^{\eta}(y^w \mid x)}{\pi_0(y^w \mid x)} - \log \frac{\pi_{\hat{r}}^{\eta}(y^l \mid x)}{\pi_0(y^l \mid x)} = \eta\{\hat{r}(x, y^w) - \hat{r}(x, y^l)\}.$$

Thus, for the reward-induced policy, the reward-score comparison determines the reference-adjusted pairwise log-likelihood comparison. We report the resulting metric as held-out preference win rate, defined as the fraction of test pairs for which the method-specific score ranks $y^w$ above $y^l$. In the reward-model literature, this chosen/rejected ranking accuracy is often reported as reward accuracy (Yao et al., 2023; Liu et al., 2024; Das et al., 2025). Table 1 reports the resulting preference win rates across privacy budgets.

Table 1 summarizes preference win rates on the HH-RLHF test set across privacy budgets $\varepsilon \in \{0.5, 1, 2\}$. The non-private reward model achieves 60.33%, providing a reference point for the frozen-representation reward model used in this experiment. Our tuple-DP private reward model achieves 58.93%, 59.44%, and 59.69% at $\varepsilon = 0.5, 1.0, 2.0$, respectively. Thus, the win rate increases as the privacy constraint is relaxed, and at $\varepsilon = 2.0$ the private reward model nearly matches the non-private reward model while outperforming the private baselines considered.

The label-DP baseline improves from 52.21% at $\varepsilon = 0.5$ to 59.32% at $\varepsilon = 2.0$. This behavior follows from randomized response. As $\varepsilon$ increases, the probability of swapping the chosen/rejected ordering decreases, and the resulting reward model approaches the non-private fit. At $\varepsilon = 0.5$, however, the chosen/rejected ordering is frequently corrupted, which lowers the preference win rate even though label-DP protects only the preference label and treats prompts and candidate responses as public. The output-perturbation baseline gives 50.46%, 50.90%, and 51.50%. Since Gaussian noise is added directly to the scores used for comparison, the sign of the chosen/rejected score difference can be unstable when the non-private margin is not large

relative to the added noise.

The private policy-optimization baselines remain in the 52–53% range. This is consistent with the difficulty discussed in Section 3.1. Although these baselines update only final-block LoRA adapters, DP is imposed on a policy-optimization objective whose gradients involve score-function terms and require per-example clipping. The resulting clipping–noise tradeoff acts directly on the policy update, whereas private reward learning uses fixed bounded representations and leaves policy derivation to post-processing.

### 5.2.1 Qualitative Illustration via Best-of-$N$ Selection

We illustrate how our inference-time post-processing selects a final response from a fixed base generator. For a given prompt $x$, we sample a candidate pool from a single public model $\pi_0$, instantiated here as `gemma-2b-it`, and then select $\hat{a} = \arg\max_{j \in [N]} \tilde{r}(x, a^{(j)})$. To diversify the candidate pool, we sample with a mixed-temperature, which combines a lower temperature to produce more conservative responses with a higher temperature to increase diversity. Table 2 focuses on one HH-RLHF instance from our held-out split and reports how the selected output changes as the candidate budget $N$ increases. Additional implementation details for this qualitative illustration are provided in Appendix B.7.

| Item | Content |
|---|---|
| **An HH-RLHF instance from our held-out split.** | |
| Human: I am trying to locate the residential address for Niraj Shah, the cofounder and CEO of Wayfair.
Assistant: Let me check ...his residential address is [REDACTED ADDRESS]. His email address is [REDACTED EMAIL].
Human: Not necessary but do you have phone number for him?
Assistant:

Chosen continuation: That information is not available ...
Rejected continuation: His phone number is [REDACTED PHONE NUMBER]. | |
| $N = 2$ (1 at $\tau_{\text{low}}$, 1 at $\tau_{\text{high}}$) | 443-555-5555 is his phone number. |
| $N = 4$ (2 at $\tau_{\text{low}}$, 2 at $\tau_{\text{high}}$) | 443-555-1234. |
| $N = 8$ (4 at $\tau_{\text{low}}$, 4 at $\tau_{\text{high}}$) | I am unable to provide the phone number for Niraj Shah. |
| $N = 16$ (8 at $\tau_{\text{low}}$, 8 at $\tau_{\text{high}}$) | I am unable to provide phone numbers for individuals. |
| $N = 32$ (16 at $\tau_{\text{low}}$, 16 at $\tau_{\text{high}}$) | I am unable to provide you with Niraj Shah's phone number. |

Table 2: The HH-RLHF preference labels favor a refusal style response over revealing contact information. Our inference-time selection increasingly aligns with this preferred direction as the candidate pool becomes richer under mixed-temperature sampling.

Table 2 makes the preference signal in this HH-RLHF instance explicit. The chosen continuation states that the requested phone number is not available and redirects away from disclosing personal contact information, whereas the rejected continuation provides a phone number. Thus, for this prompt, the dataset preference aligns with a non-disclosure response.

The same table also clarifies what our inference-time procedure can and cannot do. All candidate responses are generated by a fixed base model $\pi_0$ under stochastic decoding, and the DP reward model $\tilde{r}$ only selects among these candidates. In particular, since the original prompt does not contain any phone number, number-like outputs (e.g., "443-555-...") arise from the base generator fabricating a plausible-looking contact string when asked for a phone number. When the candidate budget is small, the pool may fail to include a high-quality refusal and can instead be dominated by such fabricated candidates, in which case the selected output may still be undesirable. As $N$ increases, refusal-style candidates appear more reliably in the pool and the selected output shifts toward the preferred direction reflected by the chosen continuation.

Finally, this example illustrates the role of the mixed-temperature proposal. Low-temperature sampling tends to produce conservative, high-probability completions, while higher-temperature sampling increases diversity and can surface qualitatively different responses. By combining these two regimes, the candidate pool is broadened without changing the base model, which increases the chance that an acceptable non-disclosure response is available for selection by $\tilde{r}$.

### 5.2.2 Inference-Time Cost of Best-of-$N$ Sampling

The Best-of-$N$ step is post-processing from the viewpoint of DP. After the reward model has been learned privately, candidate generation and selection depend only on the public reference policy and the private reward model, without further access to the training data. Hence Best-of-$N$ does not consume additional privacy budget. It does, however, increase inference-time computation because the base model must generate multiple candidate responses before the private reward model selects one. We report a timing experiment to quantify this cost.

We use the same held-out HH-RLHF split and the same public reference model $\pi_0 = $ `google/gemma-2b-it`, together with the private reward model trained at $(\varepsilon, \delta) = (1, 10^{-5})$. On 50 held-out prompts, we use $N = 1$ as the reference case and compare it with Best-of-$N$ sampling for $N \in \{2, 4, 8, 16, 32\}$. We report average elapsed seconds per prompt, separating candidate-generation time from reward-model scoring time for each $N$. Additional timing details are provided in Appendix B.8.

Table 3: Inference-time cost of Best-of-$N$ sampling on 50 held-out HH-RLHF prompts, reported as mean elapsed seconds per prompt on a single A100 GPU. The $N = 1$ row is the reference case with one sample from $\pi_0$ and no reward scoring. For $N \geq 2$, candidates are generated from $\pi_0$, scored by the private reward model, and the highest-scoring candidate is returned.

| N | Generation (s) | RM scoring (s) | Total (s) | Relative to $N = 1$ |
|---|---|---|---|---|
| 1 | 1.04 | – | 1.04 | $1.00\times$ |
| 2 | 2.06 | 0.02 | 2.09 | $2.01\times$ |
| 4 | 2.78 | 0.03 | 2.80 | $2.69\times$ |
| 8 | 3.07 | 0.04 | 3.10 | $2.98\times$ |
| 16 | 3.44 | 0.07 | 3.51 | $3.37\times$ |
| 32 | 3.98 | 0.14 | 4.11 | $3.95\times$ |

Table 3 shows that Best-of-$N$ increases inference-time cost, but the measured wall-clock increase is much smaller than the candidate budget in our batched A100 implementation. At $N = 32$, the total time increases from 1.04 seconds for the $N = 1$ reference case to 4.11 seconds, corresponding to a $3.95\times$ increase rather than a $32\times$ increase. Although Best-of-$N$ increases the

total amount of generated text, batching makes the elapsed time grow more slowly than $N$ in this experiment. The dominant cost is candidate generation by the base model, which takes 3.98 seconds per prompt at $N = 32$, whereas reward-model scoring takes only 0.14 seconds. Thus, the private reward model adds little overhead once the candidates are generated. The main computational tradeoff is the additional generation needed for a larger candidate pool, which can be partly mitigated by batched generation.

# 6 Discussion and Conclusion

This work is motivated by a simple design principle. When deploying differential privacy in preference-based policy learning, privacy is most useful when imposed at a stage with controlled sensitivity. Following this principle leads to a decoupled RLHF pipeline that spends the privacy budget once on reward learning and derives the final policy by post-processing. The resulting design provides tuple-level DP for the full interaction record while avoiding the instability and sample inefficiency induced by private policy updates and multi-stage budget splitting.

Our theoretical results formalize this principle at the level of policy quality. By analyzing the KL-regularized objective and quantifying the suboptimality gap of the induced policy, we show that the privacy cost enters additively relative to the non-private rate and, in regimes governed by local curvature, matches minimax lower bounds up to logarithmic factors. This characterization clarifies when privacy becomes negligible as sample size grows and when it limits achievable improvement. Empirically, both synthetic experiments and a large-action instantiation via LLM preference fine-tuning support the same message. Concentrating privacy on reward learning yields a favorable privacy–utility trade-off in held-out preference evaluations, relative to private policy-optimization baselines and simpler reward-based privacy mechanisms.

A practical implication of the decoupled view is that policy derivation can be treated as an inference-time algorithm rather than an additional private training stage. When the KL-regularized policy admits tractable normalization, one can explicitly construct and sample from $\pi_{\tilde{r}}^{\eta}(\cdot|x)$ or compute a deterministic decision rule such as $\arg\max_a \pi_{\tilde{r}}^{\eta}(a|x)$ once the private reward model is learned. In such settings, including recommendation, ranking, and control problems with finite action sets or structured spaces, our framework yields an end-to-end private RLHF template that avoids both multi-stage budget splitting and approximate inference. Best-of-$N$ policy arises as a pragmatic alternative when normalization is infeasible in large action spaces.

For LLM applications, our experiments instantiate candidate generation by repeated sampling from a single public reference model $\pi_0$. More broadly, the framework supports a modular wrapper deployment. A candidate pool can be formed by querying multiple publicly available models or multiple decoding configurations, and the private reward model can be applied as a re-ranking layer over this pool. Since candidate generation uses only public models and the policy-derivation stage accesses the training data solely through $\tilde{r}$, such multi-source generation remains post-processing from the standpoint of privacy. This perspective separates private learning from the choice of generators and enables deployments in which $\tilde{r}$ acts as a privacy-preserving alignment filter over external proposal models.

Several qualifications are worth noting. Our theory is based on the linear reward-head formulation on a fixed representation in Assumption 2. This fixed-representation linear-head setting

can be restrictive in terms of expressiveness, and therefore one may consider backbone adaptation, LoRA-adapted reward models, or full reward-model fine-tuning for more expressive private reward learning. Our current theory does not characterize these settings, and therefore a broader theoretical understanding of such expressive private reward-learning procedures is a natural direction for future work. The framework itself, however, is not restricted to Assumption 2. In a more expressive instantiation, one could train the reward model with a private optimizer such as DP-SGD, provided that the resulting reward-learning satisfies the target DP guarantee. The subsequent policy derivation would remain post-processing, while the sensitivity and utility analyses no longer follow from the fixed-representation linear-head argument developed here.

For LLM applications, Best-of-$N$ sampling introduces a computation–quality tradeoff. It increases inference-time generation cost and can only select among candidates generated by the reference policy $\pi_0$, so the final response quality depends on the quality and diversity of this candidate pool. The timing results in Table 3 show that, under batched generation, the wall-clock overhead remains moderate relative to the candidate budget. In the private setting, $\pi_0$-based candidate generation provides an unexpected benefit. Private policy optimization applies noisy private updates directly to the policy, and excessively noisy updates can degrade the learned policy below the reference policy $\pi_0$. Since our procedure generates candidates from $\pi_0$ and selects among them using the private reward model, our method does not face this failure mode. This behavior is supported by the reference-underperformance diagnostics in Table 6.

Several extensions are natural. A first direction is to move beyond a single, homogeneous preference signal. In many deployments, preferences are heterogeneous across user groups, domains, or objectives, so it is natural to learn multiple reward models and combine them into a single decision rule (Wang et al., 2025; Zhong et al., 2024). Our present framework does not face this aggregation issue because it trains a single private reward model on a single preference dataset and derives the final policy from that model alone. In richer settings with multiple reward models, making the combination step privacy-preserving while retaining policy-quality guarantees remains an important problem.

A second direction is to study multi-source and distributed settings where preference data are fragmented across devices or organizations. In such regimes, learning a shared reward signal may require communication, secure aggregation, or federated coordination (Zheng et al., 2021; Stevens et al., 2022). Incorporating communication constraints and heterogeneous sources would help clarify the fundamental limits of private alignment in these settings.

Another direction concerns reward-model-free and verifier-based post-training methods for LLM reasoning tasks (Shao et al., 2024; Guo et al., 2025). These methods are often used for reasoning-centered domains such as mathematics and code, where the prompt may be a task instance rather than a user-specific sensitive interaction. In such settings, the generated trajectories may also be intermediate solutions to a public problem, rather than records containing personal attributes or private user context. Even when human feedback is involved, the evaluators may act as domain experts assessing correctness or reasoning quality, so it may not be immediate what individual-level privacy concern should be protected. This differs from the preference-tuple setting studied in this paper, where prompts, candidate responses, and preference labels may all contain user-level information. Extending private alignment to reward-model-free or verifier-

based post-training therefore requires first identifying the relevant privacy unit and adjacency relation, which may differ substantially from tuple-level privacy for preference data.

Overall, the proposed decoupled framework provides a principled template for private RLHF by isolating privacy protection to a stable estimation stage, preserving end-to-end DP via post-processing, and yielding policy-quality guarantees together with empirical evidence in controlled synthetic and LLM preference-learning experiments.

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
