# OpenReview forum: "Privacy-Preserving Reinforcement Learning from Human Feedback via Decoupled Reward Modeling"
_SLADS/Section_C — Decision pending for SLADS_Section_C_

### Review · Reviewer_kayb · 2026-05-25

**Summary Of Contributions:**

This paper tackles the critical issue of tuple-level privacy leakage during the preference-based fine-tuning stage (such as RLHF) of Large Language Models. The authors identify a fundamental bottleneck in existing private alignment approaches: injecting Differential Privacy (DP) directly into policy optimization (e.g., DP-DPO or DP-PPO) leads to severe utility degradation. This happens because policy score functions are theoretically unbounded, forcing these algorithms to rely on aggressive and destructive gradient clipping to enforce privacy constraints.

The core innovation of this work is a decoupled private RLHF framework. Instead of fighting the unbounded gradients in the policy stage, the authors exploit the two-stage structure of RLHF by shifting the DP mechanism entirely to the reward modeling phase. By restricting the private learning to a linear head on top of a frozen pre-trained backbone, the input features are naturally bounded. This allows the framework to apply calibrated noise without destructive clipping, treating the downstream policy derivation (such as Best-of-N sampling) as a mathematically guaranteed, zero-cost privacy post-processing step. Key Contributions are:

1. Methodological Paradigm Shift: The authors introduce a structurally aware pipeline that completely avoids unstable private policy optimization, elegantly decoupling the privacy-preserving mechanism from the final policy generation to protect the full interaction tuple.

2. Natural Sensitivity Control: By utilizing a frozen representation with a linear reward head, the per-example gradients are inherently bounded by the feature norm ($||\phi(x,a)||_2 \le L$). This successfully eliminates the optimization bias typically caused by standard DP-SGD gradient clipping.Rigorous

3. Theoretical Foundation: The submission establishes a robust suboptimality gap upper bound of $\tilde{O}(\frac{\eta d}{n} + \frac{\eta d}{n^2 \epsilon^2})$ for the KL-regularized policy. Furthermore, it derives the first minimax lower bound for this specific setting, clearly mapping the phase transitions to prove the method is rate-optimal in the privacy-dominated regime.

**Audience:**

Yes

**Broader Impact Concerns:**

No concerns.

**Claims And Evidence:**

Yes

**Requested Changes:**

Requested Changes:
1. The proposed decoupled framework heavily relies on a critical assumption: the existence of a distinct, parameterized reward model training phase where the DP budget can be spent. However, the current frontier of LLM alignment is rapidly shifting toward RM-free paradigms, such as Group Relative Policy Optimization (GRPO) utilizing rule-based or verifier-based rewards.

2. Comparison with Simpler Baselines: Please include a theoretical or empirical comparison against simpler mechanisms like Label-DP (perturbing preference labels $y_i$) and Output Perturbation (adding noise directly to scalar rewards) to quantitatively justify the necessity of your approach over these computationally cheaper alternatives.

Strengthening:
1. The linear head restricts model expressivity, and Best-of-N sampling introduces high inference latency while bottlenecking quality on the base model's diversity. Please add a discussion acknowledging these practical limits and exploring potential pathways for active feature adaptation without breaking gradient bounds.

2. To provide a complete and transparent picture of the framework's computational footprint, please explicitly list the number of trainable parameters for your proposed method versus the baselines. Furthermore, quantitatively report the inference latency and time required for the Best-of-N sampling compared to direct generation to accurately reflect the real-world deployment costs.

**Strengths And Weaknesses:**

## Strengths:
1. Interesting Design: Leveraging the post-processing property of DP by isolating noise strictly to the reward model is a simple yet highly effective idea. It avoids the gradient clipping issues of policy optimization in the policy-training process.

2. Theoretical Rigor: The theoretical foundation is solid. The paper provides a clear suboptimality gap upper bound and introduces a novel minimax lower bound that rigorously characterizes the statistical and privacy-dominated phase transitions.

## Weaknesses:

1. High Inference Overhead (Best-of-N): Because the framework strictly avoids updating the policy parameters and relies entirely on Best-of-N (BoN) sampling, it drastically multiplies the computational cost and latency at inference time. The inference cost is essentially scaled by $N$ compared to standard direct generation methods.

2. To maintain bounded sensitivity, the method relies on a frozen backbone with a linear reward head. This restricts the model's expressivity, more complex alignment tasks may require active feature adaptation.

3. Restrictive Coverage Assumption: Assumption 4 (uniformly bounded density ratio between candidate and reference policies) is overly strong. In high-dimensional LLM action spaces, this ratio easily explodes if the target distribution diverges slightly from the SFT reference. The authors should discuss the implications of this assumption breaking in real-world text generation.

---

> ### Author Response · Authors · 2026-06-09
> **Response to Reviewer kayb, Part 1**
>
> We thank the reviewer for the careful comments. This first part addresses the inference-time cost of Best-of-$N$, candidate-pool dependence, trainable parameter counts, and the computational footprint. We address expressivity, coverage, reward-model-free post-training, and simpler privacy baselines in Part 2.
>
> **Inference-time cost and candidate-pool dependence of Best-of-$N$.** This addresses Weakness 1, the Best-of-$N$ latency part of Strengthening 1, and the inference-cost part of Strengthening 2. We agree that Best-of-$N$ introduces an inference-time computation cost. Following the reviewer’s comment, we added a timing experiment on 50 held-out HH-RLHF prompts and revised the Discussion to state this cost explicitly.
>
> The $N=1$ reference case takes $1.04$ seconds per prompt. Best-of-$N$ takes $2.09$, $2.80$, $3.10$, $3.51$, and $4.11$ seconds per prompt for $N=2,4,8,16,32$, respectively. Thus, at $N=32$, the measured wall-clock time is $3.95\times$ the $N=1$ case rather than $32\times$ in our batched A100 implementation. The main cost is candidate generation from the base model; reward-model scoring takes only $0.14$ seconds at $N=32$. We added these results to the new Best-of-$N$ latency subsection and summarized the computation--quality tradeoff in the Discussion.
>
> We also revised the Discussion to make the candidate-pool limitation explicit. Best-of-$N$ can only select among candidates generated by the reference policy $\pi_0$, so final response quality depends on the quality and diversity of this pool. Although this may initially make $\pi_0$-based generation appear restrictive, we also added discussion on its privacy-specific advantage. If private policy updates are too noisy, the learned policy can underperform the reference policy $\pi_0$. Our procedure instead keeps generation anchored to $\pi_0$ and uses the private reward model only for post-processing selection. This point is supported by the reference-underperformance diagnostics in Table 6 of the supplement.
>
> **Trainable parameters and computational footprint.** This addresses the parameter-count part of Strengthening 2. We revised the manuscript to report the trainable parameter counts for our method and the baselines. For reward-model methods, including our method, Label-DP, and output perturbation, the trainable component is the reward head. The reward head trains $2{,}048$ parameters and is bias-free because a global intercept cancels in the Bradley--Terry pairwise loss and does not affect pairwise reward comparisons.
>
> For private policy-optimization baselines, the trainable component is the policy adapter. All pretrained backbone parameters are frozen, and LoRA adapters are trained only in the query, key, value, and output projection modules of the final transformer block. This gives $102{,}400$ trainable parameters.
>
> A fully parameter-matched comparison is difficult because the methods optimize different objects: reward-model methods train a reward-scoring component, while DP-DPO and DP-RLHF train policy adapters. Nevertheless, the implementations share the same high-level design principle. The pretrained backbone is fixed across methods, and private training is restricted to the terminal method-specific component. We revised the LLM experiment and appendix accordingly, making the parameter counts, trained components, shared frozen-backbone design, and terminal-component difference explicit.

---

> ### Author Response · Authors · 2026-06-09
> **Response to Reviewer kayb, Part 2**
>
> This is Part 2 of our response to Reviewer kayb. It addresses expressivity, coverage, reward-model-free post-training, simpler privacy baselines, and the remaining active-feature-adaptation part of Strengthening 1.
>
> **Expressivity of the fixed-representation reward model.** This addresses Weakness 2 and the remaining expressivity/active-feature-adaptation part of Strengthening 1. We agree that using a fixed representation $\phi(x,a)$ and a linear reward head limits expressivity, since the representation layers are not adapted during reward learning. We made this limitation explicit in the Discussion.
>
> At the same time, the proposed framework itself is not restricted to a fixed-representation linear reward model. It requires a private reward-learning stage followed by policy derivation as post-processing. In principle, the private reward model could use a richer architecture, including backbone adaptation or full reward-model fine-tuning, as long as reward learning satisfies the target DP guarantee. DP-SGD can still enforce DP through per-example clipping and noise addition, but the resulting sensitivity and utility behavior would require separate analysis. We focus on the fixed-representation linear model because it is standard in theoretical RLHF analyses [1] and aligns with lightweight practice where a pretrained backbone is used as a representation map and only a limited downstream component is trained. We clarified that the current guarantees do not cover backbone-adaptive or nonlinear reward models, while the framework can still be instantiated with richer private reward-learning procedures.
>
> **Coverage assumption and implications for LLM generation.** This addresses Weakness 3. We revised the discussion of Assumption 4 to clarify its role and limitation. Assumption 4 imposes a uniform density-ratio bound between candidate policies and $\pi_0$, ruling out policies that place substantial mass on responses assigned negligible probability by $\pi_0$. This restriction is tied to the data-generation model because preference data are collected from candidates sampled under $\pi_0$, and therefore provide essentially no information about response regions that $\pi_0$ almost never generates. In other words, a policy that is essentially unrelated to $\pi_0$ cannot be learned reliably from preference data collected under $\pi_0$. Coverage assumptions formalize this limitation and are standard in offline RL and RLHF theory [2]. In LLM preference fine-tuning, where $\pi_0$ is typically an SFT or instruction-tuned model, the guarantee applies to policies whose responses remain within regions supported by $\pi_0$. If a target policy moves substantial mass outside this support, preference data collected under $\pi_0$ cannot justify reliable guarantees for those regions.
>
> **Reward-model-free and verifier-based post-training methods.** This addresses Requested Change 1. We added a discussion of reward-model-free and verifier-based post-training methods. The revised manuscript explains that these methods raise a different privacy question from the preference-tuple setting studied in our paper. The scope of our data is tuple-level preference data: prompts, candidate responses, and preference labels may all contain user-level information. In contrast, verifier-based or rule-based methods are often used in reasoning-centered domains such as mathematics and code, where prompts may be task instances and generated trajectories may be intermediate solutions to public problems. Thus, the relevant privacy unit and adjacency relation may differ substantially from tuple-level privacy for preference data. We now state this as an important direction for future work.
>
> **Simpler privacy baselines: Label-DP and output perturbation.** This addresses Requested Change 2. Following the reviewer’s suggestion, we added two simpler privacy baselines. Label-DP protects only the preference label and improves from $52.21%$ to $59.32%$ as $\varepsilon$ increases. Output perturbation clips reward scores, adds calibrated Gaussian noise, and remains near chance, from $50.46%$ to $51.50%$. Our method achieves $58.93%$--$59.69%$, close to the non-private reward model at $60.33%$. These results quantitatively compare our method with computationally simpler reward-based privacy mechanisms.
>
> References:
>
> [1] Zhu et al. *Principled reinforcement learning with human feedback from pairwise or k-wise comparisons*. International Conference on Machine Learning, 2023.
>
> [2] Song et al. *The importance of online data: Understanding preference fine-tuning via coverage*. The Thirty-eighth Annual Conference on Neural Information Processing Systems, 2024.

---

### Review · Reviewer_j9MF · 2026-05-28

**Summary Of Contributions:**

The paper studies differentially private reinforcement learning from human feedback (DP-RLHF), motivated by the fact that preference data used in alignment may contain sensitive user information. Its main contribution is a new privacy-aware RLHF pipeline that places differential privacy only on the reward-learning stage, rather than privatizing the entire downstream alignment procedure or the final policy update. After learning a private reward model, the method derives the final policy through post-processing, either by using the reward-induced optimal policy corresponding to KL-regularized RLHF/DPO-style objectives, or by using the private reward model to perform best-of-N selection among responses sampled from a reference policy. Theoretical and empirical studies are conducted to verify the results.

**Audience:**

Yes

**Broader Impact Concerns:**

No significant broader impact concerns.

**Claims And Evidence:**

Yes

**Requested Changes:**

1. Clarify the scope of the theoretical guarantees. The main theory assumes a linear reward model on fixed representations, together with bounded features, realizability, coverage, and Bradley--Terry preference modeling assumptions. The DP-SGD anlysis depends on the strong convexity. However, it is well known that the BT model is not strongly convex or even identifiable without constraint. The paper should clarify why the strongly convexity can be used in the analysis.

2. Clarify the relationship between the proposed method and DPO. The paper should be more precise about when reward-model learning followed by policy extraction differs from DPO-style optimization. In particular, if the policy class is parameterized as
$$\pi_\theta(a \mid x) \propto \pi_0(a \mid x)\exp\left(\eta r_\theta(x,a)\right).$$
then the DPO preference likelihood can reduce to a Bradley--Terry likelihood over reward differences, since the normalization terms cancel in the pairwise comparison. Under such a matched parameterization, the proposed reward-learning approach and DPO-style policy optimization may be very closely related. The authors should either formalize this connection or clearly explain why the proposed method is distinct under the model classes used in the theory and experiments.

3. In 5.1.2, I recommend adding a matched baseline in which DP-DPO uses the same reward-induced policy class as the proposed method, or the same linear feature representation and parameter dimension. This would help isolate whether the empirical advantage comes from the decoupled privacy design itself, rather than from using a simpler or better-conditioned model class.

4. Use a shared downstream evaluation metric in the LLM experiment. The LLM evaluation is somewhat indirect because the proposed method is evaluated by reward accuracy, while the policy baselines are evaluated by the win rate. The authors should add a common evaluation protocol for all methods.

5. The proposed method trains a lightweight private reward head, whereas DP-DPO and DP-RLHF update LoRA policy parameters. The paper should make the counts of trainable parameters comparable.

6. The LLM instantiation relies on selecting among candidates generated from the reference model. The method cannot improve beyond the support and quality of the candidate pool. The theoretical properties are not well studied. The paper should discuss these theoretical properties and discuss sensitivity to the candidate budget N.

**Strengths And Weaknesses:**

Strengths:

1. The paper addresses an important and timely problem. Preference data used in RLHF may contain sensitive information. The paper’s focus on protecting the full interaction tuple is therefore well motivated.

2. The main methodological idea is clean and conceptually appealing. Rather than adding DP noise at every stage of an existing RLHF or DPO pipeline, the paper proposes learning a DP reward model and then deriving the downstream policy through post-processing. This is a natural and elegant use of the post-processing property of differential privacy.

Weaknesses:

1. The theoretical analysis relies on a restricted reward model class. In particular, the main results assume a linear reward model on fixed representations. While this assumption makes the analysis tractable, it is substantially stronger than what is typically assumed in RLHF, where reward models are often nonlinear and jointly adapted with the underlying policy.

2. The connection between the proposed reward-based policy and DPO could be clarified. The policy-reward relation used in DPO implicitly assumes a sufficiently rich policy or reward class. When the reward class is restricted to a linear model over fixed features, this imposes a strong modeling assumption and may not correspond to the standard DPO formulation over a flexible policy class.

3. The comparison in Section 5.1.2 may not be fair. With an appropriately specified policy class, DPO can closely approximate the proposed method. For example, if the target policy is parameterized as a ratio adjustment of the reference policy, then the DPO loss reduces to a Bradley--Terry likelihood over reward differences. Under such a parameterization, DPO and reward-model learning followed by policy extraction would be closely related, or possibly equivalent. Thus, the empirical advantage in this section may partly reflect differences in model parameterization rather than only the benefit of the proposed privacy design.

4. The LLM evaluation is somewhat indirect. The proposed method outputs a reward model and is evaluated using reward accuracy, whereas the policy-optimization baselines are evaluated using response-only log-likelihood win rate.

5. The baseline comparison may not fully isolate the effect of the privacy design. The proposed method trains a lightweight private reward head, while the DP-DPO and DP-RLHF baselines update policy parameters through LoRA. As a result, it is difficult to determine how much of the observed performance gap is due to the proposed method versus differences in parameterization and optimization stability.

---

> ### Author Response · Authors · 2026-06-09
> **Response to Reviewer j9MF, Part 1**
>
> We thank the reviewer for the careful comments. This first part addresses the scope of the theory, the use of strong convexity in the Bradley--Terry model, and the relationship between our framework and DPO. We address the evaluation protocol, trainable parameter counts, and Best-of-$N$ implementation in Part 2.
>
> **Scope of the theory and strong convexity of Bradley--Terry reward learning.** This addresses Weakness 1 and Requested Change 1. We revised the manuscript to clarify that the linearity assumption applies only to the final reward head on top of a fixed pretrained representation $\phi(x,a)$, not to the full prompt--response map. Thus, the full input-to-reward relationship need not be linear; nonlinear structure may be encoded in the fixed representation. This fixed-representation formulation is standard in theoretical analyses of RLHF and preference-based policy learning [1,2], and is also aligned with lightweight adaptation practice where the pretrained backbone is fixed and only a downstream component is trained [3,4]. We made this scope clearer in the discussion following Assumption 2.
>
> We also clarified that we do not assume that an unrestricted Bradley--Terry model is strongly convex. The Bradley--Terry likelihood depends on $\theta$ only through pairwise reward differences, equivalently through $\langle\theta,\Delta\phi(x;a,a')\rangle$. Hence directions orthogonal to the pairwise-difference features are not identifiable from preference comparisons. Assumption 3 rules out such null directions by requiring the population pairwise-difference Gram matrix to have a positive minimum eigenvalue. Together with bounded $\Theta$ and bounded $\phi$, this gives a uniform lower bound on the logistic curvature. A matrix concentration argument then yields high-probability strong convexity of the empirical negative log-likelihood on the identified pairwise-difference parameterization. We made this explicit in the discussion following Assumption 3 and in the proof sketch of Lemma 7. This also explains why the strongly-convex DP-SGD utility guarantee is invoked only after establishing this high-probability curvature event, rather than assumed directly for the Bradley--Terry loss.
>
> **Relationship between DPO and Bradley--Terry reward learning.** This addresses Weaknesses 2--3 and Requested Changes 2--3. We revised the manuscript to explicitly acknowledge the reviewer’s point that, under a matched reward-induced policy class, DPO and Bradley--Terry reward learning are closely related. If the policy is parameterized as
> $$
> \pi_r^\eta(a\mid x)\propto \pi_0(a\mid x)\exp{\eta r(x,a)},
> $$
> then the normalizing term cancels in pairwise comparisons, and the DPO pairwise logit coincides with the Bradley--Terry reward difference up to the $1/\eta$ scaling [5]. Thus, our distinction is not the non-private change of variables itself.
>
> The distinction is what object is privatized after this relationship is recognized. DPO uses the reward-policy identity to make the policy the private training target. Our framework instead privately learns the reward model and derives the final policy by post-processing. This distinction matters under DP because private policy optimization exposes score-function gradients directly to clipping and DP noise, whereas the reward-head formulation gives transparent sensitivity control under bounded fixed representations. We added this clarification in the DPO derivation in the Preliminaries section and in the DP-DPO comparison paragraph in the Proposed Method section, connecting the reward-policy identity to the clipping--noise issue illustrated by the existing DPO/PPO examples.
>
> Regarding the suggested matched DP-DPO baseline, we agree that it is useful for interpretation. However, if DP-DPO is constrained to the same reward-induced policy class with the same fixed representation and reward parameter $\theta$, the resulting objective is essentially the Bradley--Terry reward-difference learning problem written in policy coordinates. Such a baseline would not isolate private policy optimization as a distinct mechanism. To address the concern more directly, we added reward-model baselines that keep the reward-head architecture fixed while changing the privacy mechanism: Label-DP and output perturbation.
>
> References:
> [1] Zhu et al. *Principled reinforcement learning with human feedback from pairwise or k-wise comparisons*. International Conference on Machine Learning, 2023.
>
> [2] Zhong et al. *Provable multi-party reinforcement learning with diverse human feedback*. arXiv preprint arXiv:2403.05006, 2024.
>
> [3] Evci et al. *Head2toe: Utilizing intermediate representations for better transfer learning*. International Conference on Machine Learning, 2022.
>
> [4] Hu et al. *Lora: Low-rank adaptation of large language models.* ICLR, 2022.
>
> [5] Rafailov et al. *Direct preference optimization: Your language model is secretly a reward model*. Advances in Neural Information Processing Systems, 2024.

---

> ### Author Response · Authors · 2026-06-09
> **Response to Reviewer j9MF, Part 2**
>
> This is the second part of our response to Reviewer j9MF. It addresses the held-out evaluation protocol, trainable parameter counts, and Best-of-$N$ implementation.
>
> **Held-out pairwise evaluation and method-specific scoring.** This addresses Weakness 4 and Requested Change 4. We clarified that the previous terminology could be misleading. In the original version, we used different names, such as reward accuracy for reward-model methods and win rate for policy-optimization baselines. However, these names referred to the same held-out preference-ranking event. For each held-out HH-RLHF pair $(x,y^w,y^l)$, the evaluation asks whether the method-specific score ranks $y^w$ above $y^l$. The term reward accuracy follows common reward-model evaluation practice, where the fraction of chosen/rejected pairs correctly ranked by a reward model is often reported under that name [1--3]. We therefore revised Table 1 and the surrounding text to use a single metric name, held-out preference win rate.
>
> The score source remains method-specific because the methods return different objects. For reward-model methods, including ours, Label-DP, and output perturbation, the score is the learned reward score. For policy-optimization baselines, including DP-DPO and DP-RLHF, the score is the response-only total log-likelihood under the learned policy. Thus, the score source differs, but the evaluated event is the same. For our framework, the reward-score comparison is directly connected to the induced policy: by Lemma 2, ranking $y^w$ above $y^l$ by $\hat r$ is equivalent to ranking it higher by the reference-adjusted pairwise log-likelihood of the reward-induced policy. We revised the text to make this protocol explicit.
>
> **Trainable parameter counts and parameterization differences.** This addresses Weakness 5 and Requested Change 5. We re-checked the implementation and corrected the reward-head count from $2{,}049$ to $2{,}048$. The reward head is bias-free because a global intercept cancels in the Bradley--Terry pairwise loss. The reward-head methods therefore train $2{,}048$ parameters.
>
> For private policy-optimization baselines, we freeze the pretrained backbone and train LoRA adapters only in the final transformer block, giving $102{,}400$ trainable parameters. A fully parameter-matched comparison is difficult because the methods optimize different objects. Nevertheless, the implementations share the same high-level design principle: the pretrained backbone is fixed, and private training is restricted to the method-specific terminal component. We revised the LLM experiment and appendix accordingly, clarifying the trained components, parameter counts, shared backbone/reference model/privacy target, and the terminal-component difference across methods.
>
> **Candidate-pool limitations and Best-of-$N$.** This addresses Requested Change 6. We clarified that Best-of-$N$ is an inference-time post-processing step. It can only select among candidates generated by the reference policy $\pi_0$, so final response quality depends on the quality and diversity of that pool. This is consistent with the coverage assumption. We revised the method and discussion sections to state this limitation more explicitly.
>
> We also emphasized the qualitative sensitivity to $N$ already illustrated in Section 5.2.1 in some sense. In that example, small candidate pools can select undesirable number-like outputs, while larger pools include refusal-style candidates and the selected response moves toward the preferred non-disclosure behavior. Thus, although qualitative, it shows how increasing $N$ can improve selection by enriching the candidate pool.
>
> We further added an inference-time timing experiment. The $N=1$ reference case takes $1.04$ seconds per prompt, while Best-of-$N$ takes $2.09$ seconds at $N=2$ and $4.11$ seconds at $N=32$ in our batched A100 implementation. At $N=32$, wall-clock time is $3.95\times$ the $N=1$ case rather than $32\times$, and reward-model scoring takes only $0.14$ seconds. We added these results to the Best-of-$N$ latency subsection and Discussion.
>
> Finally, we expanded the Discussion to emphasize a privacy-specific advantage. If private policy updates are too noisy, the learned policy can underperform the reference policy $\pi_0$. Our procedure keeps generation anchored to $\pi_0$ and uses the private reward model only for post-processing selection. This is supported by the reference-underperformance diagnostics, Table~6 in the supplement.
>
> References:
>
> [1] Yao et al. *Deepspeed-chat: Easy, fast and affordable rlhf training of chatgpt-like models at all scales*. arXiv preprint arXiv:2308.01320, 2023.
>
> [2] Liu et al. *Dual active learning for reinforcement learning from human feedback*. arXiv preprint arXiv:2410.02504, 2024.
>
> [3] Das et al. *Active preference optimization for sample efficient rlhf*. Joint European Conference on Machine Learning and Knowledge Discovery in Databases, 2025.

---

### Review · Reviewer_7Chr · 2026-05-28

**Summary Of Contributions:**

The paper studies Privacy-Preserving RLHF under a tuple-level privacy, rather than protecting only the preference label. The main contribution is a decoupled DP framework for RLHF that imposes differential privacy only on reward learning and derives the policy from the private reward model via post-processing. Empirically, the paper verifies this framework through a synthetic experiment and an LLM experiment.

**Audience:**

Yes

**Broader Impact Concerns:**

No concerns.

**Claims And Evidence:**

Yes

**Requested Changes:**

1. The reward model trains only a linear head with a frozen backbone. However, common practice in existing work is to fine-tune the reward model via LoRA or full-parameter tuning.
2. Table 1 only evaluates in-distribution performance (held-out prompts), but prompts used in practice are often out-of-distribution. To measure the reward model's generalization performance, common practice in prior work is to report the accuracy or other metrics on a public reward benchmark (e.g., RewardBench).
3. Section 5.1.2 lacks a fair evaluation framework. The data should be split into a training set and a held-out set. I suggest that all methods should be evaluated as follows: (1) proposed method: a tuple-level DP reward model, used to select responses via Best-of-N on the held-out prompts. (2) Label-DP RM + Best-of-N: another reward model baseline that protects only the labels, but performs Best-of-N selection on the same held-out prompts. (3) DP-DPO: a private policy trained on the training split and used to generate on the held-out prompts.
(4) DP-RLHF: a DP reward model trained on one half of the training split and a DP-PPO / private policy update trained on the other half, used to generate on the held-out prompts. Finally, the generated responses should then be evaluated by an external judge such as a strong reward model, LLM-as-a-judge, or human annotations. This design ensures a fair and reliable assessment.

**Strengths And Weaknesses:**

Strengths:
1. Protecting privacy on reward learning rather than policy optimization is a clean idea. It uses the two-stage RLHF instead of directly adding DP noise to policy optimization. This method avoids biased gradients induced by DP clipping and achieves tuple-level privacy without extra privacy cost at the policy stage.
2. On the theory side, the paper analyzes the suboptimality gap of the policy and establishes the first minimax lower bound for private RLHF problem.

Weaknesses:
1. The architecture of the reward model is simple, training only a linear head on a frozen backbone. While this aligns with the linear-reward assumption in the theory section, it limits the model performance and diverges from standard implementations of reward models in existing works.
2. The LLM experiment in section 5.2 has some issues.  First, it lacks relevant baselines: since the main contribution is the decoupled reward model, this approach should compare against other reward model–based methods (e.g., Label-DP RM) rather than only policy-optimization baselines. Second, the experimental setup in Section 5.1.2 lacks a fair comparison and a shared evaluation metric across all baselines.

---

> ### Author Response · Authors · 2026-06-09
> **Response to Reviewer 7Chr**
>
> We thank the reviewer for the careful comments. We revised the manuscript to clarify the scope of the fixed-representation reward model, add reward-model-based baselines, and make the held-out evaluation protocol more explicit.
>
> **Scope of the fixed-representation linear reward model.** This addresses Weakness 1 and Requested Change 1. We revised the manuscript to clarify that the linearity assumption applies only to the final reward head on top of a fixed pretrained representation $\phi(x,a)$, not to the full prompt--response map. Thus, the full input-to-reward relationship need not be linear; nonlinear structure may be encoded in the fixed representation. This fixed-representation linear model formulation is standard in theoretical analyses of RLHF and preference-based policy learning [1], and is aligned with lightweight adaptation practice where the pretrained backbone is fixed and only a downstream component is trained [2]. In our LLM instantiation, we follow this structure by freezing the pretrained backbone and training the reward head on the final hidden representation. We made this scope clearer in the discussion following Assumption 2.
>
> We also agree that this design can be restrictive. More expressive reward models, including backbone-adaptive or nonlinear reward models, would require separate sensitivity and utility analyses because the representation-learning step becomes data-dependent. We now state this limitation explicitly in the Discussion.
>
> **Reward-based baselines and held-out pairwise evaluation.** This addresses Weakness 2 and Requested Changes 2--3. We revised the LLM experiment to include additional reward-model baselines. In addition to the non-private reward-model reference, we added Label-DP and output perturbation. Label-DP applies randomized response to the binary preference label and therefore protects only the preference label. Output perturbation trains a non-private reward model, clips scalar reward scores, and adds Gaussian noise. These baselines complement the policy-optimization baselines by comparing privacy mechanisms within reward-model-based evaluation.
>
> Regarding the evaluation metric, we clarified that the previous terminology could be misleading. In the original version, we used different names, reward accuracy for reward-model methods and win rate for policy-optimization baselines. The term reward accuracy follows from existing reward-learning literature [3,4]. However, in our experiment, both names referred to the same held-out preference-ranking event. For each held-out test pair $(x,y^w,y^l)$, the evaluation asks whether the method-specific score ranks $y^w$ above $y^l$. We therefore revised Table 1 and the surrounding text to use a single metric name, held-out preference win rate, for all methods.
>
> The score source remains method-specific because the methods return different objects. For reward-model methods, including ours, Label-DP, and output perturbation, the score is the learned reward score. For policy-optimization baselines, including DP-DPO and DP-RLHF, the score is the response-only total log-likelihood under the learned policy. Thus, all rows evaluate the same held-out chosen/rejected ranking event, while using the natural score associated with the object returned by each method.
>
> Under the revised evaluation, the non-private reward model achieves $60.33%$. Our tuple-DP private reward model achieves $58.93%$, $59.44%$, and $59.69%$ at $\varepsilon=0.5,1.0,2.0$, respectively. Label-DP achieves $52.21%$, $55.89%$, and $59.32%$, while output perturbation achieves $50.46%$, $50.90%$, and $51.50%$. These additions clarify the reward-model privacy comparison.
>
> Regarding the suggestion to use an out-of-distribution reward benchmark such as RewardBench or an output-level generation benchmark with an external judge, we agree that such evaluations are valuable. In this revision, we focused on the controlled comparison needed to address the main Table 1 concerns: the same HH-RLHF data source, train/test split, reference model, and privacy budgets across methods, together with the missing reward-model baselines and the clarified held-out preference win-rate evaluation.
>
> References:
>
> [1] Zhu et al. *Principled reinforcement learning with human feedback from pairwise or k-wise comparisons*. International Conference on Machine Learning, 2023.
>
> [2] Hu et al. *Lora: Low-rank adaptation of large language models.* ICLR, 2022.
>
> [3] Yao et al. *Deepspeed-chat: Easy, fast and affordable rlhf training of chatgpt-like models at all scales*. arXiv preprint arXiv:2308.01320, 2023.
>
> [4] Liu et al. *Dual active learning for reinforcement learning from human feedback*. arXiv preprint arXiv:2410.02504, 2024.

---

### Decision · Action_Editor_Q3D2 · 2026-07-12

**Recommendation:** Accept as is

**Audience:**

Overall, the paper offers an elegant privacy-aware redesign of the RLHF pipeline, supported by meaningful theoretical results and substantially improved experiments. All three reviewers recommend Leaning Accept after considering the revision. I therefore recommend Leaning Accept.

**Claims And Evidence:**

This paper proposes a privacy-preserving RLHF framework that applies tuple-level differential privacy only during reward-model learning and derives the downstream policy through post-processing. The central idea is simple and compelling: by privatizing a reward model with controlled sensitivity, the method avoids the clipping and noise issues associated with private policy optimization, while incurring no additional privacy cost during policy derivation. The paper complements this methodological contribution with upper and minimax lower bounds on policy suboptimality, including a characterization of the statistical and privacy-dominated regimes, as well as synthetic and LLM experiments.

The reviewers agreed that the problem is important and the main claims are supported. Their initial concerns focused on the restricted fixed-representation linear reward model, the identifiability and strong-convexity arguments for the Bradley-Terry objective, the relationship to DPO, the fairness of the empirical comparisons, and the computational and support limitations of Best-of-N inference. The authors addressed these concerns well.

Some limitations remain. The theoretical guarantees rely on a frozen representation, linear reward head, realizability, non-degeneracy, and a relatively strong coverage condition. The LLM evaluation is still based on pairwise ranking rather than an output-level generation evaluation with an external judge, and the comparison with policy-optimization baselines does not fully match parameterization or parameter count. These limitations restrict the generality and practical applicability of the current results. However, the revised paper states these qualifications clearly, and they do not undermine the claims within the stated scope.